# ROBUST DISCOVERY OF GOVERNING EQUATIONS THROUGH SYMMETRY

## ABSTRACT

Discovering governing equations of dynamical systems directly from data remains a fundamental challenge, especially under noise and data scarcity. We propose a symmetry-inspired symbolic regression (SI-SR) framework that automatically identifies intrinsic physical invariances and embeds them into a symmetry-constrained variable set, enhancing robustness and promoting sparsity. The framework combines a validation step for symmetry confirmation with symbolic regression for expressive nonlinear modelling. We evaluate SI-SR on canonical partial differential equations (PDEs) and variable-coefficient systems, with systematic comparisons against state-of-the-art baselines. Results show that leveraging symmetry reduces redundancy and enables the recovery of compact, accurate models. This establishes symmetry as a powerful inductive bias for data-driven equation discovery.

## 1 INTRODUCTION

Discovering the governing equations of dynamical systems is a long-standing challenge in physics and engineering (Sommerfeld, 1949; Debnath & Debnath, 2005; Brunton & Kutz, 2024). Recent advances in computational resources (Fan et al., 2014) and machine learning (Jordan & Mitchell, 2015) have made it feasible to infer such equations directly from data. Existing approaches span three main directions: symbolic regression for closed-form discovery (Bongard & Lipson, 2007; Xu et al., 2020; Ma et al., 2024), sparse regression for parsimonious model selection (Brunton et al., 2016; Fasel et al., 2022; Zolman et al., 2024), and deep learning frameworks for highly expressive nonlinear modelling (Petersen et al., 2021; Xu et al., 2024; Liu et al., 2024b). However, these approaches remain fragile in noisy and data-scarce settings, often producing models that lack physical interpretability.

A fundamental limitation of purely data-driven approaches is that they fit observations rather than uncovering the underlying physical laws. Consequently, they often introduce redundant terms or overly complex structures to compensate for noise and data scarcity. To move beyond curve fitting, we propose a symmetry-inspired framework for equation discovery. By enforcing physical invariances as inductive biases, our approach automatically eliminates redundant terms, adjusts model coefficients, and recovers compact governing equations. The framework integrates three components: (i) automatic identification of system symmetries, (ii) a validation mechanism to confirm the identified symmetries, and (iii) symbolic regression for expressive equation recovery.

Our main contributions are:

- **Symmetry identification:** A framework that automatically extracts and validates system symmetries directly from data, enabling discovery even when prior physical knowledge is unavailable.

- **Validation mechanism:** A new criterion that both verifies identified symmetries and serves as a stopping rule.

- **Comprehensive evaluation:** Extensive benchmarks on both constant- and variable-coefficient PDEs under diverse noise conditions, showing improved robustness and generalisation over state-of-the-art baselines.

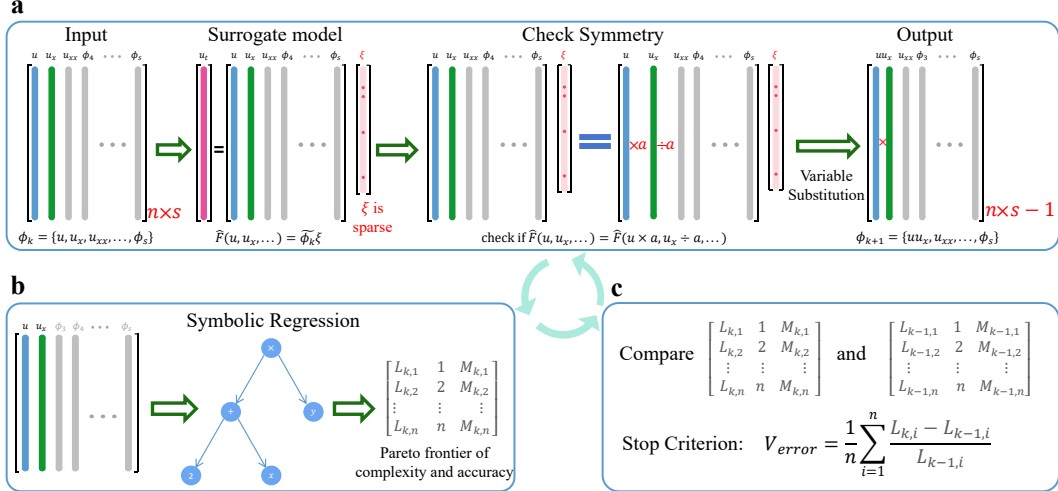

Figure 1: Schematic overview of the SI-SR framework. **a** Symmetry module reduces the variable set by eliminating redundancies. **b** Symbolic regression searches for governing equations via genetic algorithms, yielding a Pareto front that balances accuracy and complexity. **c** Validation compares successive iterations to confirm variable substitutions.

## 2 RELATED WORKS

**Approximation of derivatives.** Accurately estimating derivatives from scarce and noisy data is a central challenge in data-driven equation discovery. Weak-form formulations mitigate this issue by avoiding explicit differentiation (Gurevich et al., 2019; Tran et al., 2024; Wang et al., 2021), but they rely on carefully designed test functions, which is impractical in high-dimensional systems. Gaussian process regression offers satisfactory robustness but requires assumptions about the equation form (Raissi & Karniadakis, 2018; Raissi et al., 2017). Automatic differentiation provides another route, offering strong resilience to noise and gaining widespread adoption in deep neural networks (DNNs) (Rumelhart et al., 1986; Raissi, 2018; Chen et al., 2022) and physics-informed neural networks (PINNs) (Raissi et al., 2019; Chen et al., 2021; Zhao et al., 2025).

**Symbolic and sparse regression.** Symbolic regression, typically implemented via genetic programming (Koza, 1994; Schmidt & Lipson, 2009; Sun et al., 2024), seeks to infer governing equations by evolving candidate expressions through mutation and crossover. It is well-suited for nonlinear systems, but often suffers from overfitting and incurs high computational cost due to the exponential search space. Sparse regression (Brunton et al., 2016; Cortiella et al., 2021; Liu et al., 2024a), in contrast, assumes that only a few dominant terms govern the system, leading to a sparse coefficient vector over a predefined function library. Sparse regression strikes a balance between accuracy and interpretability, yet it is highly sensitive to noise and relies on manually specified libraries where nonlinear terms must be provided a priori.

**Symmetry-inspired methods.** Symmetry is a cornerstone of physics, with a one-to-one correspondence to conservation laws (Noether, 1971), and has been increasingly leveraged in data-driven equation discovery. For instance, embedding energy-preserving quadratic nonlinearities into sparse regression frameworks reduces overfitting and reveals intrinsic physical structures (Loiseau & Brunton, 2018). Euclidean symmetries (Reinbold et al., 2021; Gurevich et al., 2024) and Galilean symmetries (Gurevich et al., 2024; Chen et al., 2024) have also been used to constrain the function library, enabling the recovery of governing equations from noisy and limited data. These studies demonstrate that symmetry provides a principled route from data to physical laws, yielding models that are more interpretable and physically consistent. However, existing approaches typically assume the relevant invariances are known a priori, restricting their applicability to systems whose governing mechanisms are not well understood.

## 3 METHOD

SI-SR employs a recursive framework consisting of three modules (Fig. 1). In each iteration, the symmetry module uses sparse regression to identify invariances of the underlying equation within the current variable set, where derivative terms are estimated using automatic differentiation (see Appendix B.1 for details). A reduced variable set is then constructed based on the identified symmetries. Symbolic regression is applied to this updated set to recover candidate governing equations. The process repeats until the termination criterion is satisfied, yielding a Pareto front of compact models.

We consider governing equations of the form:

$$u_t = F(u, u_x, u_{xx}, \dots), \quad F \in \mathcal{F}(\phi_1 = \{u, u_x, u_{xx}, \dots\}), \tag{1}$$

where $\mathcal{F}(\phi_1)$ denotes nonlinear functions generated by the initial variable set $\phi_1$. A system is symmetric if its dynamics remain invariant under a transformation $\mathcal{T}$:

$$[\tilde{x}, \tilde{t}, \tilde{u}] = \mathcal{T}(x, t, u), \quad \tilde{u}_{\tilde{t}} = \mathcal{T}(F)(\tilde{u}, \tilde{u}_{\tilde{x}}, \tilde{u}_{\tilde{x}\tilde{x}}, \dots), \quad F = \mathcal{T}(F). \tag{2}$$

Joint discovery can be formulated by augmenting the data loss with a residual enforcing symmetry invariance:

$$\min_{F \in \mathcal{F}(\phi_1)} ||u_t - F|| + \lambda ||F - \mathcal{T}(F)||. \tag{3}$$

However, this is intractable since $\mathcal{T}$ may belong to an infinite class of transformations. Our key idea is to shift the search from $\mathcal{T}$ to the variable set $\phi$, reformulating the problem as:

$$\min_{F \in \mathcal{F}(\phi), \, \phi} ||u_t - F|| + \lambda ||\phi - \mathcal{T}(\phi)||, \tag{4}$$

which enforces symmetry through the choice of variables. This allows us to decouple the task into two recursive subproblems: (i) symmetry identification via variable selection, and (ii) governing equation recovery via symbolic regression.

### 3.1 SYMMETRY-INSPIRED METHOD

**Identification of symmetry.** For symmetric systems, $F$ depends only on the local invariants derived from the variable set (see Appendix A for the theoretical proof). This can often be represented as dependence on variable pairs (Udrescu & Tegmark, 2020), e.g.,

$$F(u, u_x, u_{xx}, \dots) = H(u \odot u_x, \, u_{xx}, \dots), \tag{5}$$

where $\odot \in \{+, -, \times, \div\}$. The optimisation problem then reduces to

$$\min_{F \in \mathcal{F}(\phi_2)} |u_t - F|, \quad \phi_2 = \{u \odot u_x, \, u_{xx}, \dots\}. \tag{6}$$

which eliminates redundant terms in $\mathcal{F}(\phi_1)$ and mitigates overfitting. For instance, $\odot = \times$ corresponds to Galilean symmetry. Applying this procedure across all variable pairs enables identification of broader invariances.

To test whether $F$ satisfies equation 5, we define symmetry errors:

$$\epsilon_{1,2,+} = \mathrm{median} |\hat{F}(u, u_x, \cdots) - \hat{F}(u + a, u_x - a, \cdots)|, \tag{7}$$

$$Error = \{\epsilon_{i,j,\odot}/p_{i,j}\}_{i<j, \, \odot \in \{+,-,\times,\div\}}, \tag{8}$$

where $p_{i,j}$ is the second-smallest among the four errors and $\hat{F}$ is an interpolation of $F$. This reduces symmetry identification to testing $F$'s dependence on variable pairs, making the problem tractable from data. The use of $p_{i,j}$ as a normalising factor improves numerical robustness; further discussion is provided in Appendix F.

**Interpolation of $F$.** Evaluating the symmetry requires estimating $F$ at locations not included in the original dataset. To this end, we employ sparse regression as a surrogate model, leveraging the

assumption that the dynamics are governed by only a few dominant terms. Under this assumption, equation 1 can be approximated as:

$$u_t = \tilde{\phi}_1 \xi, \quad \tilde{\phi}_1 = \{1, \phi_1, \phi_1^2, \dots\}, \tag{9}$$

where $\tilde{\phi}_1$ denotes an extended library with polynomial terms, and $\xi$ is a sparse coefficient vector.

The regression problem is formulated as:

$$\xi = \arg\min_{\xi} \|u_t - \tilde{\phi}(u)\xi\|_2^2 + \lambda\|\xi\|_0, \tag{10}$$

with $\lambda$ controlling sparsity. Since solving (10) exactly is NP-hard, we adopt sequential threshold Lasso (STL) as an efficient approximation (Algorithm S2).

## 3.2 SYMBOLIC REGRESSION

Symbolic regression (Cranmer, 2023) seeks interpretable models by combining operators and variables to form candidate expressions for $F$. The operator library includes basic arithmetic and common analytic functions:

$$\mathcal{O} = \{+, -, \times, \div, \sin, \ln, \exp, \dots\}. \tag{11}$$

While expressive, applying symbolic regression directly to the initial variable set $\phi_1$ often results in overfitting and low accuracy, particularly under noise or data scarcity.

**Model selection.** At each variable set $\phi_k$, symbolic regression produces a Pareto front of candidate models $\{L_{k,i}, i, M_{k,i}\}_{i=1}^n$, where $L_{k,i}$ is the error, $i$ is the complexity, and $M_{k,i}$ is the symbolic expression. We select the final model using a score-based criterion that balances accuracy and complexity:

$$M_i < M_{i+p} \quad \text{if} \quad \frac{L_i - L_{i+p}}{pL_i} > \frac{0.1}{d}, \tag{12}$$

where $M_i < M_{i+p}$ indicates that model $M_{i+p}$ is superior to $M_i$, and $d$ is the spatial dimensionality. This criterion evaluates the improvement in accuracy relative to added complexity, while incorporating a global scaling factor. A model is *pre-optimal* if it is not dominated by any other, and the final choice is the simplest among all pre-optimal models.

## 3.3 VALIDATION CRITERION

To assess identified symmetries, we monitor symbolic regression performance across successive variable sets. A valid symmetry should reduce inaccuracy without increasing complexity, as merging variables simplifies the structure of $F$. Conversely, invalid substitutions typically yield a sharp rise in error, since the reduced set cannot represent the target dynamics. We formalise this effect with the validation error:

$$V_{\text{error}} = \frac{1}{n}\sum_{i=1}^n \frac{L_{k,i} - L_{k-1,i}}{L_{k-1,i}}, \tag{13}$$

which measures the average relative change in inaccuracy between iterations. If $V_{\text{error}} < 1$, the symmetry is deemed valid and the recursion continues; otherwise, the process terminates to avoid propagating errors.

## 4 RESULTS

**Baselines.** Since no existing baselines directly address our setting, we design ablation studies and compare against representative approaches for symmetry identification and PDE discovery:

- **Symmetry identification:** AI-Feynman (Udrescu & Tegmark, 2020), originally developed for static systems. To adapt it, derivative terms are provided via numerical estimation.
- **Sparse regression:** PDE-FIND (Rudy et al., 2017) and its extension to variable-coefficient PDEs (GSTR) (Rudy et al., 2019). To assess robustness under noise, we also include DL-PDE (Xu et al., 2019), which estimates derivatives using DNNs.

Table 1: Comparison of baselines in terms of physics usage, source of information, nonlinear term construction, and sparsity mechanism.

| Method | Physics info | Source | Nonlinear terms | Sparsity |
|---|---|---|---|---|
| SI-SR | ✓ | data | automatic | intrinsic |
| PDE-FIND | ✗ | ✗ | predefined | sparse regression |
| DL-PDE | ✗ | ✗ | predefined | sparse regression |
| ICNet | ✓ | prior knowledge | ✗ | intrinsic |
| Symmetric SINDy | ✓ | prior knowledge | predefined | intrinsic |

- **Symmetry-based methods:** ICNet (Chen et al., 2024) and Symmetric SINDy (Xie et al., 2022), both of which reduce the function library using prior physical knowledge.

A summary of these baselines, which compares their use of physics, source of information, nonlinear term construction, and sparsity mechanism, is summarised in Table 1.

## 4.1 BASIC BENCHMARKS

We first evaluate SI-SR on four canonical PDE systems: Burgers', Korteweg–de Vries (KdV), Navier–Stokes (NS), and FitzHugh–Nagumo reaction–diffusion (FN) equations.

**Symmetry identification.** Symmetry is critical for robustness to noise, whereas incorrect identification can yield invalid variable sets. To assess this capability, we test SI-SR's ability to detect system symmetries. As shown in Table 2, SI-SR rapidly and accurately recovers the complete set of symmetries across all benchmarks.

Table 2: Comparison with AI-Feynman on symmetry identification for basic benchmarks. Reported are the identified symmetries and the average running time (in seconds) per symmetry. Ground-truth symmetries are listed in the third column, and correctly identified symmetries are highlighted in bold. Details on the symmetry identification process can be found in Appendix E.1.

| Benchmark | Method | True symmetry | Identified symmetry | Running time (s) |
|---|---|---|---|---|
| Burgers' | SI-SR | Galilean | $(\boldsymbol{u}, \boldsymbol{u_x}, \times)$ | **0.49** |
| | AI-Feynman | | $(u_{xxx}, u_{xxxx}, \times); (u_{xx}, u_{xxx}u_{xxxx}, \times)$ | 54.2 |
| KdV | SI-SR | Galilean + scaling | $(\boldsymbol{u}, \boldsymbol{u_x}, \times)$ | **0.49** |
| | AI-Feynman | | $(u_x, u_{xxx}, \times)$ | 29.9 |
| FN | SI-SR | None | $(\boldsymbol{v}, \boldsymbol{u_{yy}}, -); (\boldsymbol{v} - \boldsymbol{u_{yy}}, \boldsymbol{u_{xx}})$ | **21.6** |
| | AI-Feynman | | $(u_x, u_{xy}); (\boldsymbol{u_{xx}}, \boldsymbol{u_{yy}}, +)$ $(\boldsymbol{v}, \boldsymbol{u_{xx}} + \boldsymbol{u_{yy}}, -)$ | 94.5 |
| NS | SI-SR | Euclidean | $(\boldsymbol{u}, \boldsymbol{\omega_x}, \times); (\boldsymbol{v}, \boldsymbol{\omega_y}, \times)$ $(\boldsymbol{u\omega_x}, \boldsymbol{v\omega_y}, +); (\boldsymbol{\omega_{xx}}, \boldsymbol{\omega_{yy}}, +)$ | **16.5** |
| | AI-Feynman | | $(\omega_y, \omega_{xx}, \div)$ | 94.0 |

**PDE discovery.** We evaluate PDE discovery using the average relative error of the identified non-zero coefficients, which directly measures model accuracy. This metric is meaningful only when the correct equation form is recovered; otherwise, if essential terms are missing or spurious terms are present, the error is reported as NaN. Table 3 summarises results across benchmarks under varying noise levels.

ICNet is restricted to Galilean symmetry and Symmetric SINDy to Euclidean symmetry, limiting their applicability to systems with these invariances. In contrast, SI-SR reformulates symmetry as dependence on variable pairs, enabling generalisation across a broader range of cases. For instance, it recovers the non-trivial composition of coordinate transformations with Galilean invariance in KdV, and detects symmetries in FN that lack a direct physical interpretation. These results demonstrate the broader applicability of our framework beyond classical symmetry assumptions.

Table 3: Average relative coefficient error on basic benchmarks under different levels of Gaussian white noise. Errors are reported only when the correct equation form is recovered; otherwise NaN indicates failure.

| Benchmark | Method | 0% | 1% | 5% | 10% | 20% |
|---|---|---|---|---|---|---|
| Burgers' | SI-SR | 0.23 | 0.43 | 0.73 | 0.97 | 1.86 |
| | PDE-FIND | NaN | NaN | NaN | NaN | NaN |
| | DL-PDE | 0.45 | 0.65 | 1.85 | 3.41 | NaN |
| | ICNet | **0.04** | **0.07** | **0.35** | **0.88** | **1.4** |
| | Symmetric SINDy | NaN | NaN | NaN | NaN | NaN |
| KdV | SI-SR | **0.07** | **0.12** | **0.24** | **0.29** | **3.1** |
| | PDE-FIND | NaN | NaN | NaN | NaN | NaN |
| | DL-PDE | 0.86 | 1.5 | 4.04 | 5.3 | NaN |
| | ICNet | NaN | NaN | NaN | NaN | NaN |
| | Symmetric SINDy | NaN | NaN | NaN | NaN | NaN |
| FN | SI-SR | **0.08** | **0.08** | **0.11** | **0.21** | **0.4** |
| | PDE-FIND | 0.74 | 2.7 | NaN | NaN | NaN |
| | DL-PDE | 1.7 | 1.9 | 2.7 | 4.8 | NaN |
| | ICNet | NaN | NaN | NaN | NaN | NaN |
| | Symmetric SINDy | 0.74 | 2.7 | NaN | NaN | NaN |
| NS | SI-SR | 1.4 | **1.3** | **1.2** | **1.4** | **1.7** |
| | PDE-FIND | 1.4 | 5.3 | NaN | NaN | NaN |
| | DL-PDE | NaN | NaN | NaN | NaN | NaN |
| | ICNet | NaN | NaN | NaN | NaN | NaN |
| | Symmetric SINDy | **0.7** | 2.2 | 7.1 | NaN | NaN |

## 4.2 ADVANCED BENCHMARKS

To evaluate SI-SR on more complex systems, we consider two representative classes: (i) $\lambda$–$\omega$ reaction–diffusion (RD) equations, and (ii) variable-coefficient PDEs, including spatially dependent advection–diffusion (VC-AD) and temporally dependent Burgers' (VC-Burgers').

### 4.2.1 REACTION–DIFFUSION EQUATIONS

Unlike the earlier benchmarks, where symmetries can be applied directly, some systems cannot be readily handled by the symmetry method. In such cases, selecting a variable set and coordinate system that preserve the equation's symmetries is crucial. However, no unified strategy exists for choosing coordinate systems and candidate functions (Li et al., 2017; Chandrashekar & Sahin, 2014).

We illustrate this challenge using the RD equations, where the symmetry method fails when applied directly. The initial variable set is:

$$\phi^u = \{u, v, u_x, u_y, u_{xx}, u_{xy}, u_{yy}\}, \tag{14}$$

in which only additive symmetry appears in the diffusion term, while the symmetries in $\lambda$ and $\omega$ are ignored. To refine the candidate set, we employ STL for variable selection, retaining only the active terms (see Appendix for details). This yields

$$\phi_1 = \{u, u^3, u^2v, uv^2, v^3, u_{xx}, u_{yy}\}. \tag{15}$$

When symmetries cannot be identified through natural variables, the coordinate system can instead be reselected, that is, the variable set redefined, to flexibly handle more complex systems. This is demonstrated in Table 4, where both AI-Feynman and SI-SR successfully identify all symmetries on the redefined variable set, with SI-SR achieving this at significantly lower running time.

### 4.2.2 VARIABLE-COEFFICIENT EQUATIONS

Variable coefficients often involve composite nonlinear terms, where traditional approaches either construct parameterised dictionaries based on expert knowledge (Goyal & Benner, 2022; Champion et al., 2020) or provide only discrete approximations (Rudy et al., 2019). To test SI-SR under this more challenging setting, we compare it with GSTR, a representative baseline for variable-coefficient discovery. The results are summarised in Fig. 2.

Table 4: Comparison with AI-Feynman on symmetry identification for RD equations and variable-coefficient Benchmarks. Details on the symmetry identification process can be found in Appendix E.2.1 and E.2.2.

| Benchmark | Mehtod | Identified symmetry | Running times |
|---|---|---|---|
| RD | SI-SR | $(u^3, uv^2, +)(u^2v, v^3, +)(u_{xx}, u_{yy}, +)$ $(u, u^3 + uv^2, -)(u - u^3 - uv^2, u^2v + v^3, +)$ | **24.3** |
| | AI-Feynman | $(u^3, uv^2, +)(u^2v, v^3, +)(u_{xx}, u_{yy}, +)$ $(u, u^3 + uv^2, -)(u - u^3 - uv^2, u^2v + v^3, +)$ | 94.2 |
| VC-AD | SI-SR | $(x, u_x, \times)(u, xu_x, +)$ | **1.18** |
| | AI-Feynman | $(t, u_{xxx}, \times)(u_{xx}, tu_{xxx}, +)$ | 28.8 |
| VC-Burgers' | SI-SR | $(u, u_x)$ | **1.01** |
| | AI-Feynman | $(u_{xx}, u_{xxx})$ | 28.1 |

The main challenge in variable-coefficient systems is that nonlinear coefficients such as $\sin(\pi t/4)$ cannot be manually constructed. SI-SR accurately recovers both active terms and their coefficients, even under 20% noise. By explicitly capturing the temporal dependence of coefficients, it also maintains high predictive accuracy beyond the training region (Fig. 2b), demonstrating strong generalisation to settings where coefficients vary over time or space.

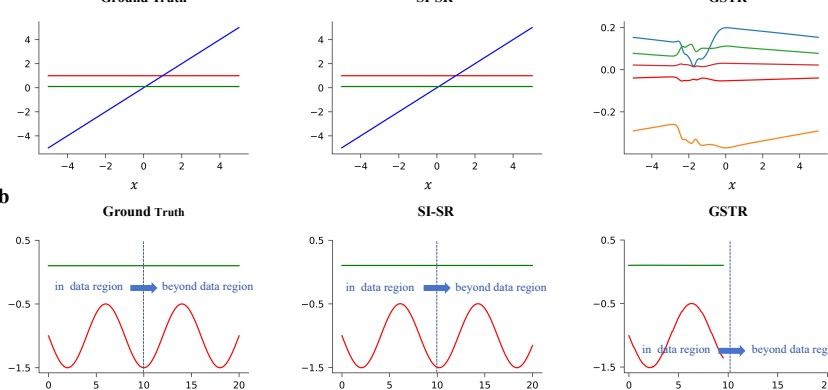

Figure 2: Comparison of SI-SR and GSTR on variable-coefficient PDEs with clean data. **a** VC-AD: SI-SR recovers the true $x$-dependence of the coefficient, while GSTR introduces spurious fluctuations. **b** VC-Burgers': SI-SR identifies the correct time dependence and extrapolates beyond the training region, whereas GSTR fails to generalise. Additional comparisons under noisy data are reported in Appendix E.2.2.

## 5 SENSITIVITY ANALYSIS

In this section, we investigate the sensitivity of SI-SR to different perturbations and settings, aiming to assess the robustness of the framework under more challenging conditions.

**Different noise types.** To evaluate robustness in practical scenarios, we test SI-SR on the NS equation under different noise distributions. We consider additive uniform noise and Gamma-distributed

noise. Unlike uniform noise, Gamma noise is strictly positive and introduces a persistent bias, which severely degrades numerical differentiation. Nevertheless, as shown in Table 5, SI-SR remains robust across both symmetric and biased noise types.

Table 5: Average relative coefficient error for the NS equation under different noise types and levels.

| Noise type | 1% | 5% | 10% | 20% |
|---|---|---|---|---|
| Uniform | 2.20 | 1.75 | 1.10 | 1.96 |
| Gamma | 2.06 | 1.49 | 1.52 | 2.33 |

**Sampling density.** We further analyse the sensitivity of SI-SR to sampling density under Gaussian noise, with results summarised in Fig. 3.

Across different spatial and temporal resolutions, SI-SR remains accurate over a wide range of spatial grids but shows stronger dependence on the number of temporal samples (Fig. 3, left). This behaviour aligns with the spectral bias of the DNN used for derivative estimation (Rahaman et al., 2019), which requires sufficiently dense temporal sampling to generalise effectively.

When jointly varying data availability and noise, SI-SR is robust under moderate sparsity and noise but deteriorates in extreme cases (Fig. 3, right). This failure arises because extreme noise obscures the derivative structure, causing the symmetry module to lock onto spurious variable dependencies.

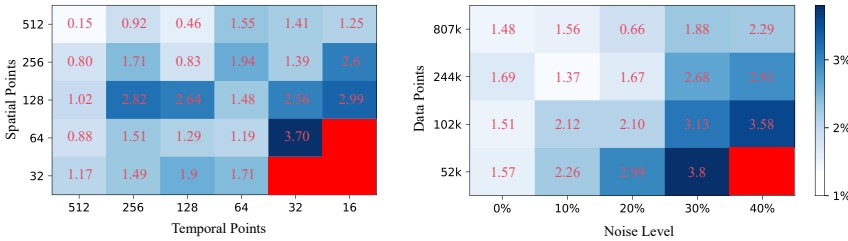

Figure 3: Sensitivity of SI-SR to sampling density and noise. Left: effect of temporal and spatial resolution on Burgers' equation, where SI-SR is more sensitive to temporal sparsity than spatial sparsity. Right: effect of data availability and Gaussian noise level on the NS equation, showing robustness under moderate noise but degradation in extreme cases. Colours indicate coefficient error.

## 6 CONCLUSION

We presented SI-SR, a symmetry-inspired framework for discovering high-dimensional PDEs with constant or variable coefficients from noisy data. Unlike purely data-driven methods, SI-SR moves beyond curve fitting by embedding physical invariances as inductive biases.

Experiments demonstrate that SI-SR achieves superior robustness and generalisation compared to state-of-the-art baselines, while remaining competitive on tasks where specialised methods excel. The framework offers two main advantages: (i) it scales effectively with data noise and scarsity, overcoming a central limitation of existing approaches; and (ii) it recovers explicit PDEs with temporally and spatially varying coefficients, maintaining accuracy even beyond the observed data domain. This combination of robustness, scalability, and explicit equation recovery highlights SI-SR as a general framework for interpretable scientific machine learning.

## 7 CODE AND DATA AVAILABLE

An anonymized repository containing all code and datasets used in this work is publicly accessible at: https://anonymous.4open.science/r/SI-SR-9E52/.

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

# A THEORETICAL PROOFS

**Theorem.** If a system is symmetric, then the governing equation can be expressed as

$$F = H(I_1, \ldots, I_m),$$

where $\{I\}_1^m$ are the local invariants.

**Proof.** Let $u : (x, t) \in \mathbb{R}^2 \to \mathbb{R}$ be a smooth map, and consider the $k$-th jet space defined by

$$J^k = \{(u, u_x, u_{xx}, \ldots, u_{(k)})\}. \tag{16}$$

Clearly, $F \in \mathcal{C}^\infty(J^k)$. Let $G$ be a smooth local Lie group acting on $u$ and its variables (with prolongation applied to derivatives). Denote by $\mathcal{T}$ the prolonged action of $G$ on the jet space $J^k$. If the equation is invariant under this group action, then for any $(x, t, u)$ satisfying the equation, the transformed field also satisfies it:

$$\forall g \in G, \ \forall j \in J^k, \quad F(\mathcal{T}_g(j)) = F(j). \tag{17}$$

If the local quotient space $J^k/G$ is a smooth manifold, then by the quotient manifold theorem there exists a natural projection

$$\pi : j \in J^k \mapsto \bar{j} \in J^k/G, \tag{18}$$

which is a smooth submersion. We can therefore define a function $H$ on the quotient space:

$$H : \bar{j} \in J^k/G \ \mapsto \ F(j) \in \mathbb{R}. \tag{19}$$

Since $F$ is invariant under the action of $G$, this definition is well posed. Finally, we obtain

$$F = H \circ \pi = H(I_1, \ldots, I_m), \tag{20}$$

where $\{I\}_1^m$ are the projected coordinates (local invariants). Moreover, many transformations also involve time, which requires a more detailed discussion. We therefore provide explicit proofs for specific symmetries in the following subsections.

## A.1 SYMMETRY

**Burgers' Equation.** The Galilean transformation is expressed as:

$$\begin{bmatrix} \tilde{x} \\ \tilde{t} \\ \tilde{u}(\tilde{x}, \tilde{t}) \end{bmatrix} = T_1(x, t, u) = \begin{bmatrix} x - ct \\ t \\ u(x, t) - c \end{bmatrix}, \tag{21}$$

where $c$ is the relative frame velocity. The state and its derivatives satisfy the following relations:

$$\frac{\partial u}{\partial t} = \frac{\partial (\tilde{u} + c)}{\partial t} = \frac{\partial \tilde{u}}{\partial \tilde{x}} \frac{\partial \tilde{x}}{\partial t} + \frac{\partial \tilde{u}}{\partial \tilde{t}} \frac{\partial \tilde{t}}{\partial t} = -c\tilde{u}_{\tilde{x}} + \tilde{u}_{\tilde{t}},$$

$$\frac{\partial u}{\partial x} = \frac{\partial (\tilde{u} + c)}{\partial t} = \frac{\partial \tilde{u}}{\partial \tilde{x}} \frac{\partial \tilde{x}}{\partial x} + \frac{\partial \tilde{u}}{\partial \tilde{t}} \frac{\partial \tilde{t}}{\partial x} = \tilde{u}_{\tilde{x}},$$

$$\cdots = \cdots$$

$$\frac{\partial^n u}{\partial x^n} = \frac{\partial}{\partial x} \left( \frac{\partial^{n-1} u}{\partial x^{n-1}} \right) = \frac{\partial}{\partial \tilde{x}} \left( \frac{\partial^{n-1} \tilde{u}}{\partial \tilde{x}^{n-1}} \right) = \frac{\partial^n \tilde{u}}{\partial \tilde{x}^n}. \tag{22}$$

Substituting the relations into Eq. equation 1 yields:

$$\tilde{u}_{\tilde{t}} - c\tilde{u}_{\tilde{x}} = F(\tilde{u} + c, \tilde{u}_{\tilde{x}}, \tilde{u}_{\tilde{x}\tilde{x}}, \cdots) = \tilde{F}(\tilde{u} + c, \tilde{u}_{\tilde{x}}, \tilde{u}_{\tilde{x}\tilde{x}}, \cdots). \tag{23}$$

This result shows that non-differential terms must be excluded from the right-hand side; otherwise, the transformed equation would not be equivalent to the original one. Furthermore, in order to balance the term $\tilde{u}_{\tilde{x}}$ appearing on the left-hand side, the right-hand side must include the nonlinear term $uu_x$. Hence the function $F$ necessarily depends on $uu_x$, and the governing equation must be defined over the symmetry-reduced set $\{uu_x, u_{xx}, \cdots\}$.

---

**Algorithm S1** Symmetry-Inspired Symbolic Regression (SI-SR)

---

**Require:** Initial variable set $\phi_1$; target derivative $u_t$; sparsity penalty coefficient $\alpha$; tolerance $tol$
1: Initialise iteration counter $k \leftarrow 0$
2: **repeat**
3:    $k \leftarrow k + 1$
4:    **Symmetry identification:** apply sparse regression to fit $\hat{F}$ and compute symmetry errors for variable pairs
5:    **Variable set reduction:** construct a reduced set $\phi_{k+1}$ using the identified symmetries
6:    **Symbolic regression:** perform symbolic regression on $\phi_{k+1}$ to obtain candidate models $\{M_{k,i}\}$
7:    **Validation:** evaluate candidate models using the validation error $V_{\text{error}}$
8: **until** $V_{\text{error}} \geq 1$    (termination criterion)

---

**KdV Equation.** The invariance of the KdV equation corresponds to a composite transformation that combines a Galilean shift with a spatial rescaling:

$$T_2 \begin{bmatrix} x \\ t \\ u(x,t) \end{bmatrix} = \begin{bmatrix} x/\sqrt{6} \\ t \\ u(x,t) \end{bmatrix}, \tag{24}$$

Under $T_2$, the governing equation is transformed into:

$$\tilde{u}_{\tilde{t}} = -\tilde{u}\tilde{u}_{\tilde{x}} - \tilde{u}_{\tilde{x}\tilde{x}\tilde{x}} \tag{25}$$

The equation is Galilean invariant, with the corresponding transformation:

$$\begin{bmatrix} \tilde{x} \\ \tilde{t} \\ \tilde{u}(\tilde{x},\tilde{t}) \end{bmatrix} = T_2 \circ T_1 \circ T_2 \begin{bmatrix} x \\ t \\ u(x,t) \end{bmatrix}, \tag{26}$$

In the SI-SR framework, such invariances are reformulated as conditions on the functional dependencies of $F$. This perspective generalises the notion of symmetry beyond classical transformation groups, enabling a more flexible characterisation of invariance in dynamical systems.

**NS equation.** A system is said to exhibit Euclidean symmetry if its governing equation remains invariant under the transformation:

$$\begin{bmatrix} \tilde{x} \\ \tilde{y} \end{bmatrix} = T_3 \begin{bmatrix} x \\ y \end{bmatrix} = \begin{bmatrix} y \\ x \end{bmatrix}, \tag{27}$$

where $T_3$ denotes a spatial coordinate flip. This invariance implies that spatially symmetric terms, such as $u\omega_x$ and $v\omega_y$, must share identical coefficients. Enforcing this symmetry within SI-SR allows the algorithm to automatically adjust the coefficients of equivalent terms and thereby eliminate redundancy in the discovered equations.

# B   DETAILS OF METHOD

The overall procedure of SI-SR is summarised in Algorithm S1, while the STL subroutine is provided separately in Algorithm S2.

## B.1   APPROXIMATION OF DERIVATIVES

In practice, only state measurements are typically available, and derivative terms must therefore be approximated numerically. To estimate derivatives, SI-SR employs a fully connected neural network, denoted by $NN_\theta = NN(x,t;\theta)$, where $\theta$ represents the trainable parameters. The network consists of an input layer, an output layer, and multiple hidden layers. Each hidden layer applies an affine transformation followed by a nonlinear activation function:

$$z_{l+1} = \sigma(W_l z_l + b_l), \tag{28}$$

---

**Algorithm S2** Sequential Threshold Lasso (STL) regression: STL($\tilde{\phi}$, $u_t$, $\alpha$, $tol$)

---

**Require:** $\tilde{\phi}$: extended function library; $u_t$: target function; $\alpha$: sparsity penalty coefficient; $tol$: minimum coefficient threshold

1: Initialise iteration counter $k \leftarrow 0$
2: **repeat**
3:    $k \leftarrow k + 1$
4:    **Sparse regression:** perform Lasso to estimate coefficients

$$\hat{\xi} = \arg\min_{\xi} \frac{1}{2n_{\text{samples}}} \left\| u_t - \tilde{\phi}\xi \right\|_2^2 + \alpha\|\xi\|_1$$

5:    **Thresholding:** identify indices of coefficients smaller than $tol$

$$\mathcal{I}_k = \{i : |\hat{\xi}_i| < tol\}$$

6:    **Enforce sparsity:** set small coefficients and corresponding terms to zero

$$\hat{\xi}_{\mathcal{I}_k} = \mathbf{0}, \quad \tilde{\phi}_{\mathcal{I}_k} = \mathbf{0}$$

7: **until** $\mathcal{I}_k = \mathcal{I}_{k-1}$

---

where $z_l$ is the output of the $l$-th layer, $W_l$ is the weight matrix, $b_l$ the bias vector, and $\sigma$ the activation function.

We adopt the sine activation function to capture periodic and self-similar behaviour while alleviating the vanishing gradient problem Hochreiter (1998). The weights $W_l$ and biases $b_l$ are initialised from a uniform distribution $\mathcal{U}(-\sqrt{k}, \sqrt{k})$, where $k$ is the dimension of the input $z_l$. Training is performed using the Adam optimiser Kingma (2014), with weight decay set to 0 and hyperparameters $\beta_1 = 0.9$, $\beta_2 = 0.999$, and $\epsilon = 10^{-8}$.

The loss function is defined as:

$$L(\theta) = \frac{1}{N_m \|\boldsymbol{D}_u\|_2} \|NN_\theta(\boldsymbol{D}_c) - \boldsymbol{D}_u\|_2, \tag{29}$$

where $\boldsymbol{D}_c$ are the coordinates of measurement points, $\boldsymbol{D}_u$ the corresponding observed values, and $N_m$ the number of samples. Once training is complete, automatic differentiation is applied to compute the required derivative terms.

The early stopping technique, summarised in Algorithm S3, is employed during neural network training to reduce computational cost and prevent overfitting. The DNN is optimised iteratively, and its performance is evaluated on a validation set every 50 epochs. If the validation loss ceases to decrease, the training process is terminated early. Since generalisation beyond the measurement domain is not required, the training data are reused as the validation set.

## C   DATASETS

**Burgers' equation.** Burgers' equation models the propagation and reflection of shock waves, as well as viscous fluid flow. It has been widely used in fluid dynamics and applied mathematics, particularly in the study of gas dynamics and hydrodynamics. The one-dimensional form is

$$u_t = -uu_x + \gamma u_{xx}, \tag{30}$$

where $\gamma$ is the diffusion coefficient (set to 0.2). The dataset is generated from the analytical solution

$$u(t, x) = -\frac{4\gamma px}{px^2 + 2\gamma pt},$$

evaluated over the domain $[2, 8] \times [-8, 8]$ with $p = 1$. The field is discretised on a uniform grid of 501 spatial points and 201 time steps. To emulate sparse measurements, 70 spatial locations are uniformly subsampled and 67 time steps are retained, corresponding to only $4.6\%$ of the full dataset.

---

**Algorithm S3** Early stopping for neural network training: $\boldsymbol{\theta} = \text{EarlyStop}(\mathbf{D}_c, \mathbf{D}_u, lr, \epsilon)$

---

**Require:** $\mathbf{D}_c$: coordinates of data points $\{t_i, x_i\}_i$; $\mathbf{D}_u$: measurement data at $\mathbf{D}_c$; $lr$: initial learning rate; $\epsilon$: tolerance threshold for early stopping

1: Initialise neural network $NN$, parameters $\boldsymbol{\theta}$, epoch counter $epoch \leftarrow 0$, loss history list $LH \leftarrow []$

2: **for** $k = 1, 2, 3, 4$ **do**

3:     Set optimiser = Adam(initial learning rate = $lr$)

4:     **Forward pass:** compute loss

$$Loss = L(\mathbf{D}_u, NN(\mathbf{D}_c)), \quad LH \leftarrow LH \cup \{Loss\}$$

5:     **Backward pass:** update parameters

$$\boldsymbol{\theta} \leftarrow \text{optimizer}(\boldsymbol{\theta}), \quad epoch \leftarrow epoch + 1$$

6:     **if** $epoch \bmod 50 = 0$ **then**

7:         Compute moving averages of recent losses

$$L_1 = \text{mean}(LH[-100 : -50]), \quad L_2 = \text{mean}(LH[-50 :])$$

8:         Evaluate decrease rate

$$dr = (L_2 - L_1)/L_1$$

9:         **if** $dr > \epsilon$ **then**

10:            Reduce learning rate: $lr \leftarrow lr/10$; **break**

11:         **end if**

12:     **end if**

13: **end for**

---

**KdV equation.** The Korteweg–de Vries (KdV) equation provides a dispersive regularisation of Burgers' equation. First derived by Boussinesq in 1877 to model shallow-water waves, it is now widely used in water wave dynamics, plasma physics, and optical fibre modelling. The KdV equation is

$$u_t = -6uu_x - u_{xxx}, \tag{31}$$

where $u$ denotes the wave amplitude. A key feature of the KdV equation is its soliton solution,

$$u(x,t) = \frac{c}{2} \operatorname{sech}^2\left[\frac{\sqrt{c}}{2}(x - ct - x_0)\right], \tag{32}$$

which demonstrated for the first time that solitary waves in nonlinear systems can be stable and persistent, profoundly influencing the development of nonlinear science and mathematical physics.

To distinguish the KdV equation from the one-way wave equation $u_t = -cu_x$, we generate time-series data for wave profiles with two initial amplitudes, $c = 2$ and $c = 6$. The spatial domain is discretised into 501 grid points and sampled at 201 time steps. To simulate sparse observations, 94 spatial locations are uniformly selected as fixed sensors, and 49 time steps are retained, corresponding to 35.9% of the dataset used in Rudy et al. (2017).

**FN equation.** The FN equation is a fundamental model in mathematical biology, widely used to describe the propagation of nerve impulses and often regarded as a prototype for excitable systems. Its form is

$$u_t = \gamma_u \nabla^2 u + u - v - u^3 + \alpha, \tag{33}$$

where $u$ and $v$ are two interacting components (e.g., biological variables), $\gamma_u$ is the diffusion coefficient (set to 1), and $\alpha$ is the coefficient for the reaction terms (set to 0.01).

For data generation, we subsample the open dataset provided in Chen et al. (2021), selecting spatial grid nodes of size $98 \times 98$ within the domain $[1, 50] \times [1, 50]$ over 101 time steps in $[7, 21]$. This corresponds to only 1.76% of the original dataset.

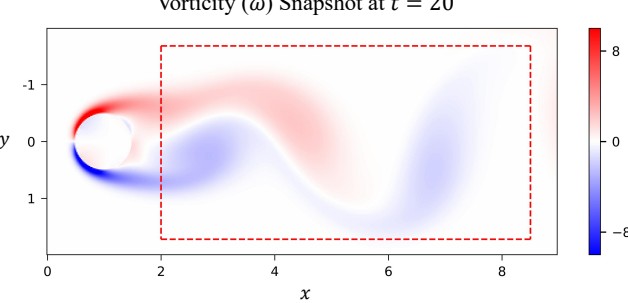

Figure S1: The data is uniformly sampled behind the cylinder in the red dashed line area.

**NS equation.** The NS vorticity equation describes the evolution of vorticity in viscous flows and plays a fundamental role in fluid mechanics. Its two-dimensional form is

$$\omega_t = -(\mathbf{u} \cdot \nabla)\omega - \gamma \nabla^2 \omega, \tag{34}$$

where $\omega$ denotes the fluid vorticity, $\mathbf{u} = \{u, v\}$ the velocity field, and $\gamma$ (set to 0.01) the kinematic viscosity.

The computational domain is discretised on a grid of $499 \times 199$ spatial points over 151 time steps. To emulate sparse measurements, we select $2,886 = 74 \times 39$ spatial locations in the wake region behind the cylinder (Fig. S1) as fixed sensors, and retain 33 time steps.

**RD equations.** The $\lambda$–$\omega$ reaction–diffusion system models the pattern formation of interacting chemical species subject to both reaction and diffusion. It has broad applications in chemical kinetics, biological morphogenesis, and population dynamics. The governing equations are

$$\begin{cases} u_t = 0.1\nabla^2 u + \lambda(g)u - \omega(g)v, \\ v_t = 0.1\nabla^2 v + \omega(g)u + \lambda(g)v, \end{cases} \tag{35}$$

where $u$ and $v$ denote the concentrations of the reacting species, and $g = \sqrt{u^2 + v^2}$. The function $\lambda(g) = 1 - g^2$ characterises the growth or decay rate, while $\omega(g) = -g^2$ specifies the angular frequency governing oscillatory behaviour.

The dataset is constructed by subsampling $103 \times 103$ spatial grid points over 99 time steps, corresponding to $7.9\%$ of the full data.

**Spatially dependent advection-diffusion equation.** The advection–diffusion equation governs the spatiotemporal transport of mass, energy, and other physical quantities, and is widely applied in fluid mechanics, heat conduction, and pollutant dispersion. Its one-dimensional form is

$$u_t = u + xu_x + \gamma u_{xx}, \tag{36}$$

where $\gamma$ is the diffusion coefficient (set to $0.1$).

The PDE is solved using a spectral method on the spatial domain $[-5, 5]$ and temporal interval $[0, 3]$, discretised with 501 spatial grid points and 201 time steps. To emulate sparse measurements, 35 spatial locations and 54 time steps are uniformly subsampled, corresponding to only $1.8\%$ of the full dataset.

**Temporally dependent Burgers' equation.** The Burgers' equation with a time-varying advection coefficient is

$$u_t = a(t)uu_x + 0.1u_{xx}, \tag{37}$$
$$a(t) = -\left(1 + \tfrac{1}{2}\sin\left(\tfrac{\pi}{4}t\right)\right). \tag{38}$$

The solution is computed using a spectral method. The spatial domain is discretised with 501 grid points, and the temporal evolution is recorded over 201 time steps under periodic boundary conditions. To emulate sparse measurements, 35 spatial locations and 54 time steps are uniformly subsampled, corresponding to only $1.8\%$ of the full dataset.

## D   HYPERPARAMETERS

Although our algorithm does not require extensive hyper-parameter tuning, several essential choices must be specified to ensure stable performance. The heuristic criteria adopted in this study are summarised below.

### D.1   HYPERPARAMETERS FOR SI-SR

**Variable and operator library.** In all numerical experiments, the operator library is defined as

$$\mathcal{O} = \{+, -, \times, \div, \text{square}, \sin, \log, \exp\}. \tag{39}$$

Since we assume minimal prior knowledge of the underlying physics, the only assumption being that the temporal evolution of $u$ can be expressed in terms of itself and its spatial derivatives, the variable set is chosen to include as many candidate terms as possible. This over-complete representation allows the symmetry identification step to eliminate redundancies systematically.

**Early stopping threshold $\epsilon$.** This hyper-parameter defines the criterion for terminating neural network training. The goal is to avoid overfitting, so $\epsilon$ is typically set to a sufficiently large value (e.g., $10^{-4}$). An exception occurs in the case of the $\lambda$–$\omega$ RD equations: as the system dynamics become more complex, the loss function may exhibit slowly oscillating descent. Premature early stopping in this case could lead to underfitting. Therefore, we set $\epsilon = -1$, indicating that training should not be stopped early.

**Tolerance $tol$.** This hyper-parameter controls sparsity in STLasso. Since STLasso is used to construct a surrogate model of the unknown function $F$, we choose a small value (e.g., $0.05$) to ensure that the identified model reflects the symmetries of the underlying equation. It should be noted that $tol = 0.05$ may not be optimal for the Navier–Stokes equations, as it can cause the symmetry module to miss invariances in $\nabla^2\omega$. However, this introduces only minor inaccuracies in coefficient estimation. In practice, expert knowledge or adaptive strategies can be employed to set $tol$ more precisely (e.g., slightly below the kinematic viscosity coefficient).

Table S1: Summary of variable sets and neural network architectures used in SI-SR. 'Variable set' lists the candidate terms included in the function library. 'Neural network' denotes the layer width configuration of the derivative-approximation network.

| Benchmarks | Variable set | Neural network |
|---|---|---|
| Burgers' | $\{u, u_x, u_{xx}, u_{xxx}, u_{xxxx}\}$ | $\{2, 128, 128, 64, 64, 1\}$ |
| KdV | $\{u, u_x, u_{xx}, u_{xxx}, u_{xxxx}\}$ | $\{2, 128, 128, 64, 64, 1\}$ |
| FN | $\{u, v, u_x, u_y, u_{xx}, u_{xy}, u_{yy}\}$ | $\{3, 128, 128, 64, 64, 1\}$ |
| NS | $\{u, v, \omega, \omega_x, \omega_y, \omega_{xx}, \omega_{xy}, \omega_{yy}\}$ | $\{3, 128, 128, 64, 64, 1\}$ |
| RD | $\{u, u^3, u^2v, uv^2, v^3, u_{xx}, u_{yy}\}$ | $\{3, 128, 128, 64, 64, 1\}$ |
| VC-AD | $\{t, x, u, u_x, u_{xx}, u_{xxx}\}$ | $\{2, 128, 128, 64, 64, 1\}$ |
| VC-Burgers' | $\{t, x, u, u_x, u_{xx}, u_{xxx}\}$ | $\{3, 128, 64, 64, 1\}$ |

**Sparsity coefficient $\alpha$.** This hyper-parameter is the coefficient of the sparse penalty term in ST-Lasso. Similar to $tol$, we use a small value (e.g., $10^{-5}$), provided that the results preserve the symmetries of the governing equation. Empirical results show that SI-SR is relatively insensitive to $\alpha$, and the same value can be applied across all benchmarks.

### D.2 HYPERPARAMETERS FOR BASELINE

For baseline methods, hyper-parameters are manually tuned under different noise levels to ensure the best possible performance. The key settings are as follows.

**Tolerance $tol$.** This parameter is crucial for PDE-FIND, DL-PDE, ICNet, and symmetric SINDy, as it directly determines the final discovered equation. For each benchmark, $tol$ is tuned independently based on its coefficients: starting from a small value, we progressively increase $tol$ until the method either recovers the ground truth or eliminates all terms.

**Numerical differentiation.** In PDE-FIND and symmetric SINDy, numerical differentiation is typically performed using finite differences. In these cases, spatial points are randomly subsampled as fixed sensors to record responses over several time steps. If finite differences fail to recover the governing equation, we attempt polynomial interpolation on uniformly subsampled data. For SGTR, we select the best-performing scheme from among finite differences, polynomial interpolation, and Fourier transform methods. For DL-PDE, we use same neural network architecture and training procedure as SI-SR for derivative approximation.

**Function library.** For methods requiring manual specification of nonlinear terms (PDE-FIND, DL-PDE, ICNet, symmetric SINDy, and GSTR), we construct a nonlinear function library from the initial variable set. This includes polynomial terms up to second order and their combinations with derivative terms. For problems that require higher-order terms—such as the RD and FN equations, polynomials of the corresponding order are also included.

## E DETAILS OF RESULTS

Gaussian noise is synthetically added to the data at each monitoring location according to

$$u_{\text{noise}} = u_{\text{clean}} + (\text{noise level}) \times \text{std}(u_{\text{clean}}) \times \mathcal{N}(0, 1), \tag{40}$$

where std denotes the standard deviation and $\mathcal{N}(0, 1)$ is the standard normal distribution with mean 0 and variance 1.

For constant-coefficient PDEs, the error of the identified coefficients is defined as

$$e = \frac{1}{\#\{\xi_i : \xi_i \neq 0\}} \sum_{\xi_i \neq 0} \frac{|\xi_i - \hat{\xi}_i|}{|\xi_i|} \times 100\%, \tag{41}$$

where $\xi$ are the ground-truth coefficients, $\hat{\xi}$ the discovered coefficients, and the subscript $i$ indexes the $i$-th element of the coefficient vector.

## E.1 BASIC BENCHMARKS

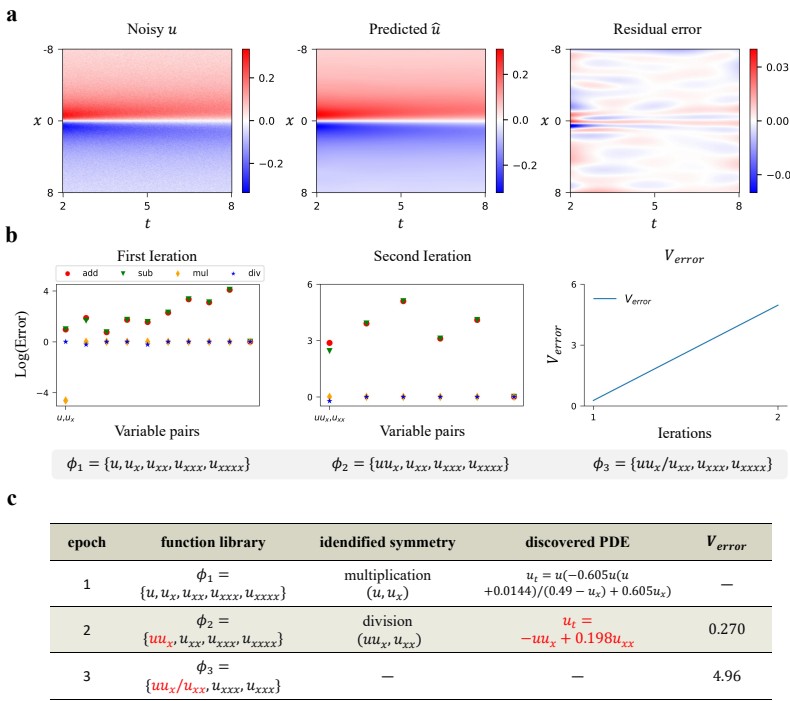

**a**

**b**

$\phi_1 = \{u, u_x, u_{xx}, u_{xxx}, u_{xxxx}\}$     $\phi_2 = \{uu_x, u_{xx}, u_{xxx}, u_{xxxx}\}$     $\phi_3 = \{uu_x/u_{xx}, u_{xxx}, u_{xxxx}\}$

**c**

| epoch | function library | idendified symmetry | discovered PDE | $V_{error}$ |
|-------|-----------------|--------------------|----------------|-------------|
| 1 | $\phi_1 = \{u, u_x, u_{xx}, u_{xxx}, u_{xxxx}\}$ | multiplication $(u, u_x)$ | $u_t = u(-0.605u(u +0.0144)/(0.49 - u_x) + 0.605u_x)$ | — |
| 2 | $\phi_2 = \{uu_x, u_{xx}, u_{xxx}, u_{xxxx}\}$ | division $(uu_x, u_{xx})$ | $u_t = -uu_x + 0.198u_{xx}$ | 0.270 |
| 3 | $\phi_3 = \{uu_x/u_{xx}, u_{xxx}, u_{xxx}\}$ | — | — | 4.96 |

Figure S2: **SI-SR for discovering the Burgers' equation with** $10\%$ **noise. a** The noisy data and predicted data, with residual error shown for comparison. The prediction is generated by the DNN, and the residual error is defined as $\mathcal{R} = \hat{u}_t - F(\hat{u}, \hat{u}_x, \ldots)$, where the hat symbol indicates values predicted by the neural network. **b** Symmetry identification across iterations. **c** Summary of each iteration, including the function library, identified symmetries, discovered PDE, and corresponding $V_{\text{error}}$.regression.

**Burgers' equation**. The results obtained from the initial variable library $\phi_1$ are summarised in Table S2 (for simplicity, only a subset is shown).

Table S2: The results of symbolic regression on $\phi_1$. The expression in bold in the table represents the optimal model.

| Inaccuracy | Complexity | Model |
|-----------|-----------|-------|
| 0.00049823 | 1 | $-0.00142$ |
| 0.00014152 | 3 | $0.0923u_{xx}$ |
| 0.00007114 | 5 | $-0.0615u + 0.0615u_{xx}$ |
| 0.00006626 | 7 | $-0.051u + 0.472u_{xx}$ |
| 0.00004352 | 9 | $u(-u^2 + 0.472u_x)$ |
| 0.00002856 | 11 | $u(-1.27u^2 + 0.472u_x)$ |
| 0.00002585 | 13 | $\mathbf{u(-0.605u(u + 0.0144)/(0.49 - u_x) + 0.605u_x)}$ |

Due to the presence of noise and redundant elements in $\phi_1$ and the operator library, direct symbolic regression suffers from overfitting. The symmetry method then exploits the invariance properties of $F$ to optimise the model. As shown in Fig. S2, the method identifies a multiplicative symmetry between $(u, u_x)$. Based on this, we replace the two variables $u$ and $u_x$ with a single new variable and

construct a symmetry-reduced function library $\phi_2 = \{uu_x, u_{xx}, u_{xxx}, u_{xxxx}\}$. Symbolic regression is subsequently applied to $\phi_2$ to discover the governing equation.

Table S3: The results of symbolic regression on $\phi_2$.

| Inaccuracy | Complexity | Model |
|---|---|---|
| 0.00032649 | 1 | 0.000216 |
| 0.00012726 | 3 | $0.09326u_{xx}$ |
| 0.00005523 | 5 | $\mathbf{-uu_x + 0.19814311u_{xx}}$ |
| 0.00005200 | 7 | $-0.824uu_x + 0.176u_{xx}$ |
| 0.00005200 | 9 | $-0.824uu_x + 0.176u_{xx}$ |

This symmetry-based reduction effectively mitigates overfitting by eliminating redundant terms involving $u$ and $u_x$. From the reduced library $\phi_2$, symbolic regression successfully recovers the closed-form Burgers' equation. The final step involves comparing the results of two consecutive iterations of symbolic regression to verify the correctness of the identified symmetry. The validation error of the current iteration is $0.27$, confirming that the substitution is valid.

Table S4: The results of symbolic regression on $\phi_3$.

| Inaccuracy | Complexity | Model |
|---|---|---|
| 0.00054576 | 1 | 0.00201 |
| 0.00042219 | 3 | $\mathbf{-0.00692u_{xxxx}}$ |
| 0.00039145 | 5 | $-0.00722u_{xxx} - 0.00722u_{xxxx}$ |
| 0.00035015 | 7 | $-0.0093u_{xx}(u_{xxxx} + u_{xxxx})/(uu_x)$ |
| 0.00032870 | 9 | $-0.0093u_{xx}(u_{xxxx} + u_{xxxx} + 0.0773)/(uu_x)$ |

The next symmetry iteration identified a division symmetry between $(uu_x, u_{xx})$ and translated the function library into $\phi_3 = \left\{ \frac{uu_x}{u_{xx}}, u_{xxx}, u_{xxxx} \right\}$, with a validation error of $4.96$. The results of symbolic regression on $\phi_3$ are reported in Table S4. Both the sharp increase in inaccuracy and the high validation error indicate that the symmetry identified in this iteration is incorrect. Consequently, the symmetry iteration is terminated, and the optimal model is selected, as shown in Table S3.

**KdV equation.** High-order partial derivatives pose significant challenges for numerical approximation from data. The SINDy framework (PDE-FIND) fails to identify the closed form of the KdV equation, even when provided with high-quality data. In contrast, symbolic regression attempts to compensate for derivative errors by introducing redundant terms, leading to a complex model: $u_t = -4.5uu_x - u_x - 0.68u_{xxx}$. By constructing a symmetry-reduced variable set $\phi_2$, SI-SR eliminates such redundant terms, since models that violate the identified symmetry properties are discarded. The resulting function library is $\phi_2 = \{uu_x, u_{xx}, u_{xxx}, u_{xxxx}\}$, and the optimal model discovered from $\phi_2$ is

$$u_t = -5.98uu_x - u_{xxx}. \tag{42}$$

SI-SR thus successfully recovers all active terms of the KdV equation and estimates their coefficients with high accuracy. This example illustrates the robustness of neural networks with early stopping in handling noise, particularly for higher-order derivatives, and highlights the role of validation in preventing false positives.

**FN equation.** Compared with the ground truth, the symmetry iteration correctly identifies all potential symmetries at the appropriate stage. Under $10\%$ noise, the discovered PDE is

$$u_t = u - v - u^3 + u_{xx} + u_{yy} + 0.0101. \tag{43}$$

The identified symmetry treats the combination $(v - u_{yy} - u_{xx})$ as a unified term, effectively adjusting the coefficient of $u_{yy}$ (from $0.78$ to $1$). More importantly, this reduction in model complexity facilitates the recovery of small coefficients, such as $\alpha$.

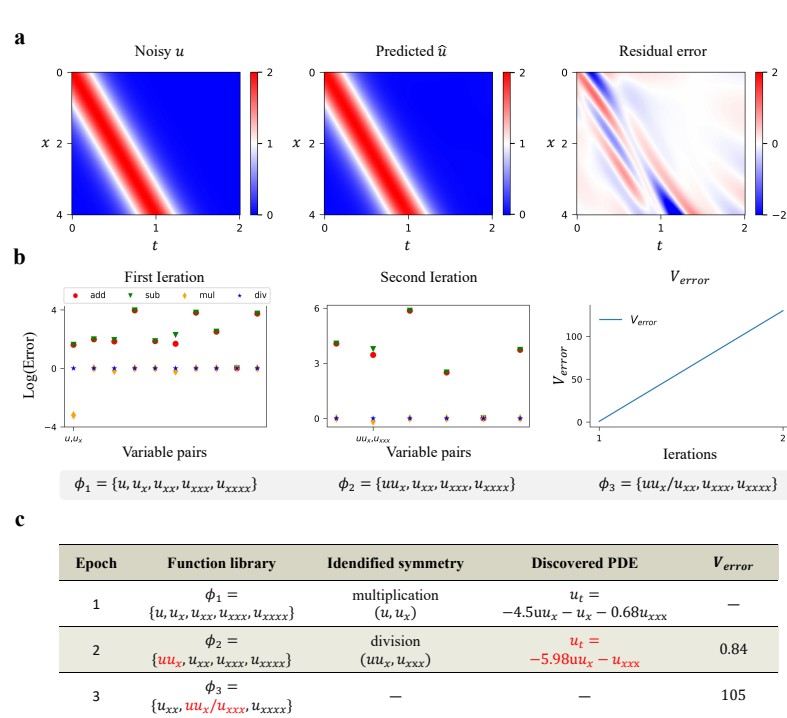

Figure S3: SI-SR for discovering the KdV equation with $10\%$ noise. **a** Noisy input data $u$, SI-SR prediction $\hat{u}$, and residual error. **b** Symmetry identification across iterations. **c** Summary of each iteration.

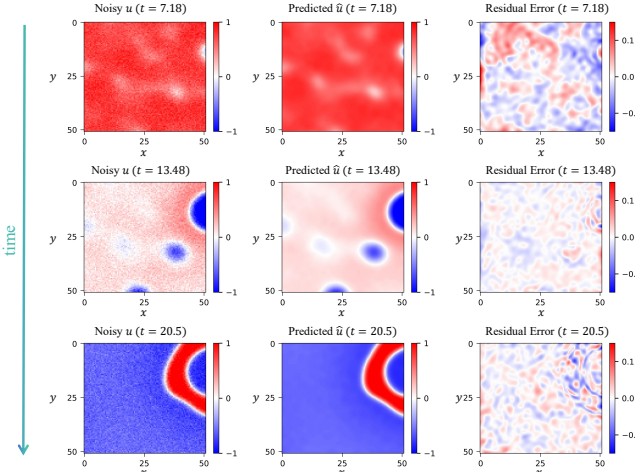

Figure S4: SI-SR results on the FN equation. Noisy input data $u$, SI-SR prediction $\hat{u}$, and residual error.

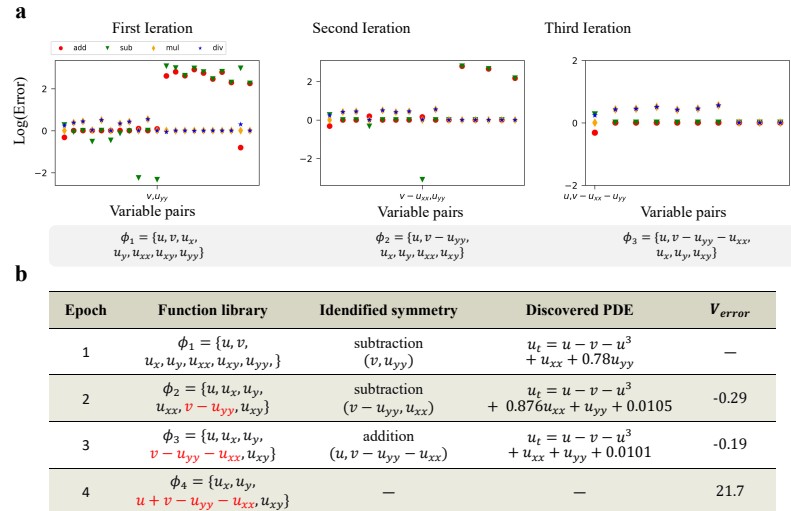

Figure S5: **SI-SR for discovering the FN equation with** $10\%$ **noise. a** Symmetry error in each variable set. **b** Symmetry identification across iterations.

It is worth noting that the coefficient $\alpha$ falls below the tolerance $tol$ and is thus removed from the surrogate model. Nevertheless, the symmetry method still identifies the underlying invariance property of the unknown function $F$, ensuring the correct discovery of the governing equation.

**NS equation.** Under $10\%$ noise, the discovered PDE is

$$\omega_t = -u\omega_x - v\omega_y - 0.01(\omega_{xx} + \omega_{yy}). \tag{44}$$

The first three identified symmetries are the most critical, as symbolic regression only converges after the third iteration. These iterations recover the advective term $(\mathbf{u} \cdot \nabla)\omega$ in the NS equation and remove all redundant related terms. The final iteration enforces the equivalence between the symmetric terms $\omega_{xx}$ and $\omega_{yy}$, ensuring they share identical coefficients.

The result is a compact model with satisfactory coefficient accuracy. This example highlights the effectiveness of SI-SR in handling complex high-dimensional systems, successfully identifying the NS equation from noisy data.

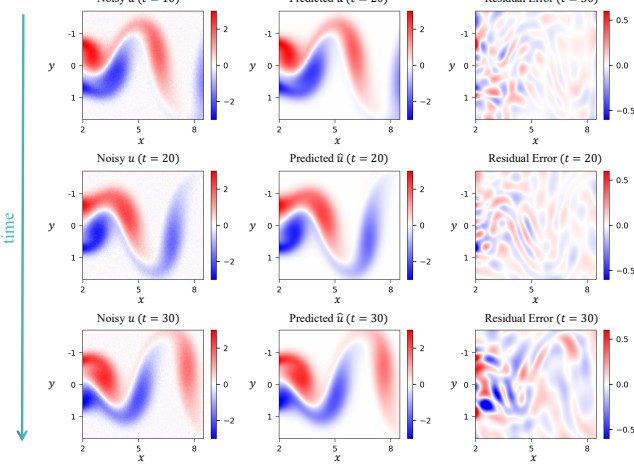

Figure S6: SI-SR results on the NS equation. Noisy input data $u$, SI-SR prediction $\hat{u}$, and residual error.

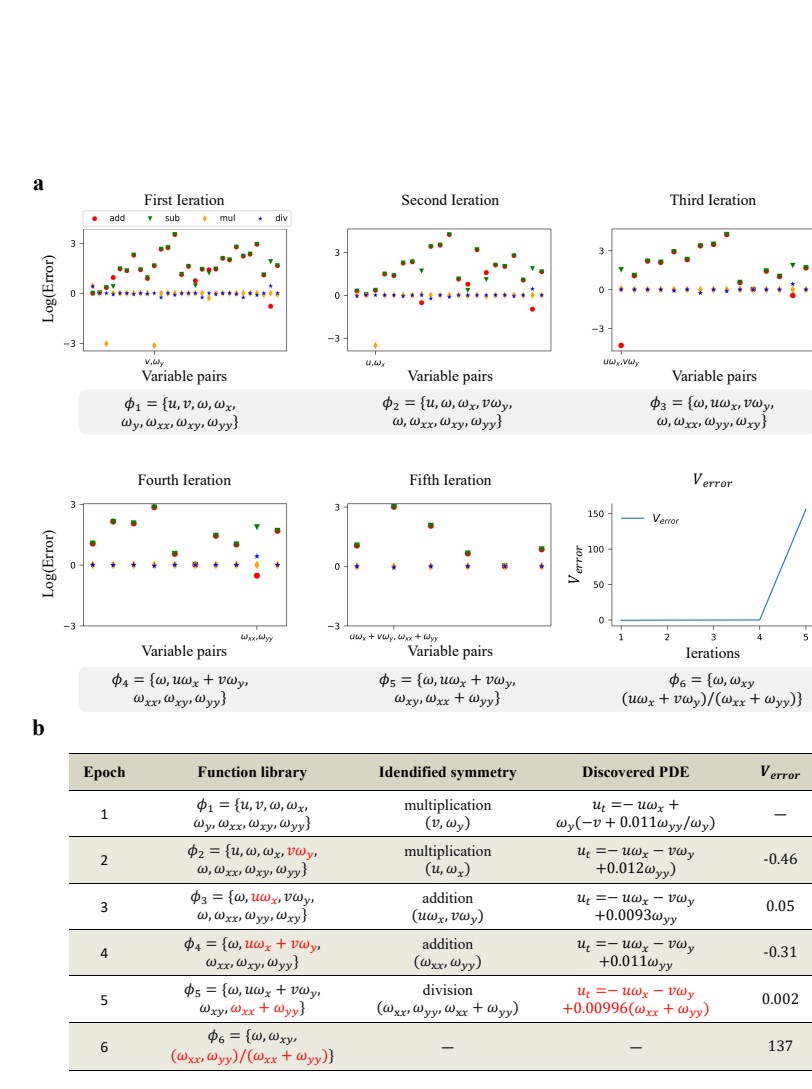

Figure S7: SI-SR for discovery of NS equation with $10\%$ noise. **a** Symmetry identification across iterations. **b** Summary of each iteration.

## E.2 ADVANCED BENCHMARKS

### E.2.1 REACTION-DIFFUSION EQUATIONS

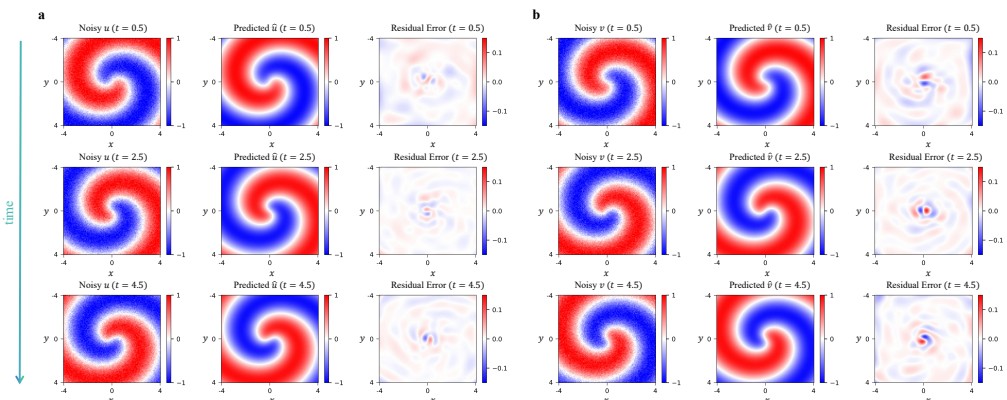

Figure S8: SI-SR results on the RD equations. **u** Noisy input data $u$, SI-SR prediction $\hat{u}$, and residual error. **v** Noisy input data $v$, SI-SR prediction $\hat{v}$, and residual error

When observing $F$ from the initial library $\phi^u$, the only apparent symmetry is the additive symmetry in the diffusion terms, while the symmetries in $\lambda$ and $\omega$ are not captured. To address this, STL is applied with $\alpha = 10^{-5}$ and $tol = 0.05$ to select candidate functions from the extended library $\tilde{\phi}^u$, which includes polynomial terms

$$\{u, v, u^2, uv, v^2\},$$

derivatives

$$\{u_x, u_y, u_{xx}, u_{xy}, u_{yy}\},$$

and their combinations. Retaining only the active terms yields the selected library

$$\phi_1^u = \{u, u^3, u^2 v, uv^2, v^3, u_{xx}, u_{yy}\}.$$

After successive symmetry iterations, the reduced library becomes

$$\phi_6 = \{u - u^3 - uv^2 + u^2 v + v^3, \ u_{xx} + u_{yy}\},$$

and the discovered equation under $10\%$ noise is

$$u_t = u - u^3 + u^2 v - uv^2 + v^3 + 0.1(u_{xx} + u_{yy}). \tag{45}$$

The symmetry iteration simplifies the governing equations, enabling accurate coefficient identification with an error of only $0.08\%$. This example demonstrates that by selecting candidate functions, the SI-SR framework can uncover hidden symmetries, thereby enhancing its applicability to complex systems. This is particularly significant because symmetry methods play a central role in physical and engineering applications.

### E.2.2 VARIABLE-COEFFICIENT EQUATIONS

**Spatially dependent advection-diffusion equation.** The identified symmetry properties enable coefficient adjustment, allowing the variable coefficients to be reconstructed with high accuracy. The results of SI-SR under different noise levels are summarised in Table S5.

**Temporally dependent Burgers' equation.** The results of SI-SR for discovering Burgers' equation under varying noise levels are summarised in Table S6. Notably, the nonlinear coefficient $\sin(\pi t/4)$ is intractable to construct manually. Nevertheless, SI-SR successfully identifies both the active terms and their coefficients with high accuracy ($\pi/4 \approx 0.785$), even in the presence of $20\%$ noise.

Moreover, by uncovering the explicit temporal dependence of the variable coefficients, SI-SR maintains high predictive accuracy beyond the data region. This represents a significant advancement for engineering applications, where long-term and large-scale measurements are often impractical, making it essential to infer global governing equations from localised samples.

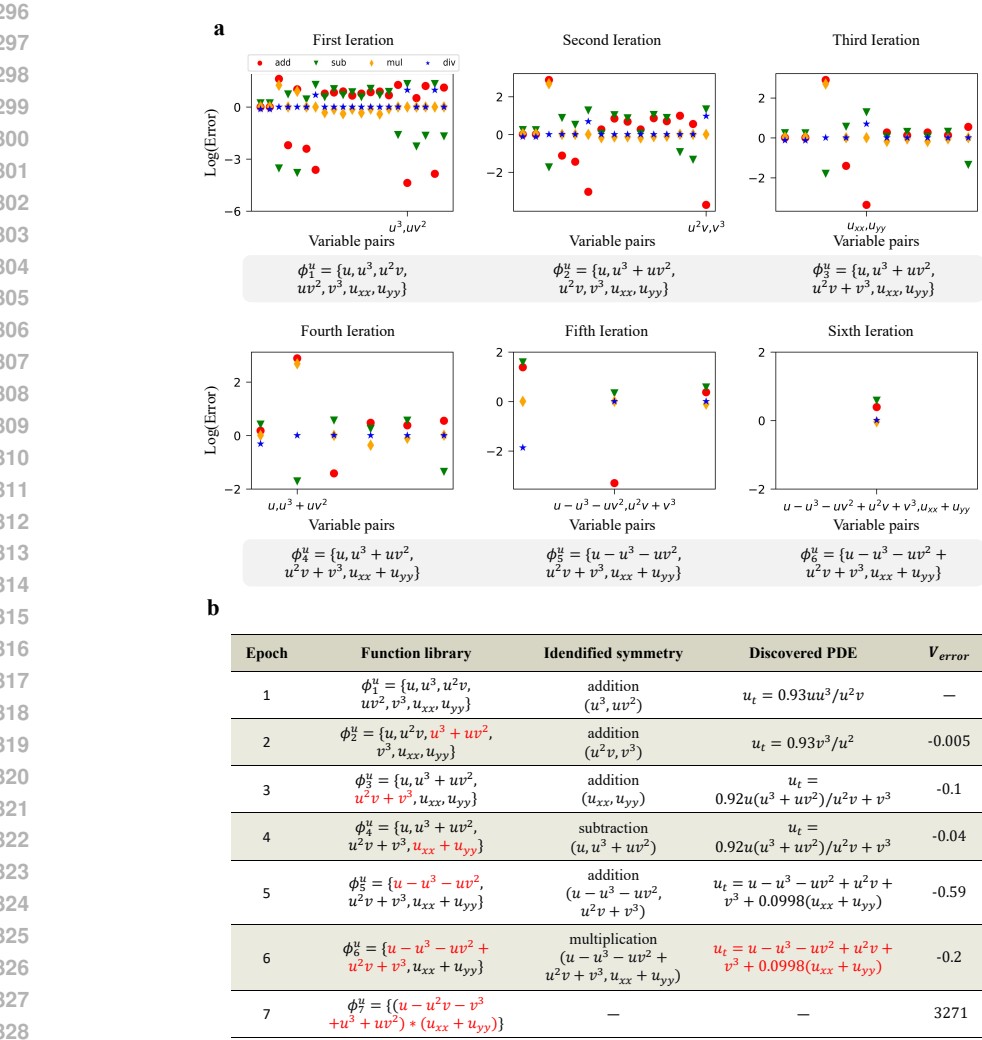

Figure S9: SI-SR for discovery of RD equations with $10\%$ noise. **a** Symmetry identification across iterations. **b** Summary of each iteration.

Table S5: Summary of SI-SR for identifying the spatially dependent Advection-Diffusion equation.

| Noise level | Discovered PDE |
|---|---|
| 0% | $u_t = u + xu_x + 0.101u_{xx}$ |
| 5% | $u_t = u + xu_x + 0.0998u_{xx}$ |
| 10% | $u_t = u + xu_x + 0.10007u_{xx}$ |
| 20% | $u_t = u + xu_x + 0.0998u_{xx}$ |

Table S6: Summary of SI-SR for identifying the temporally dependent Burgers' equation.

| Noise level | Discovered PDE |
|---|---|
| 0% | $-(1 + 0.48sin(0.788t))uu_x + 0.095u_{xx}$ |
| 5% | $-(1 + 0.48sin(0.788t))uu_x + 0.095u_{xx}$ |
| 10% | $-(1 + 0.48sin(0.788t))uu_x + 0.095u_{xx}$ |
| 20% | $-(1 + 0.45sin(0.791t))uu_x + 0.107u_{xx}$ |

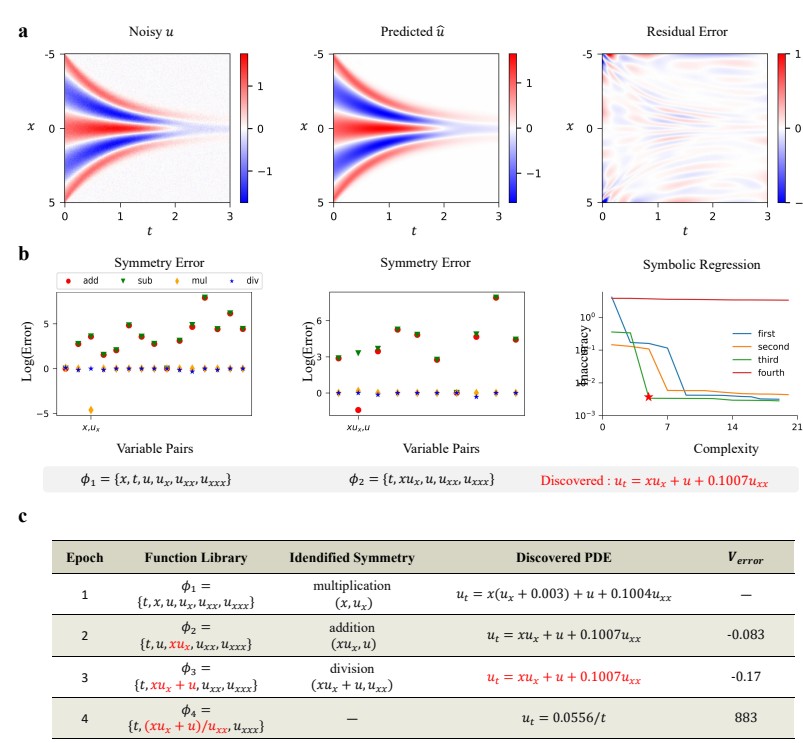

Figure S10: SI-SR results on the VC-AD equation. **a** Noisy input data $u$, SI-SR prediction $\hat{u}$, and residual error. **b** Symmetry identification across iterations. **c** Summary of each iteration

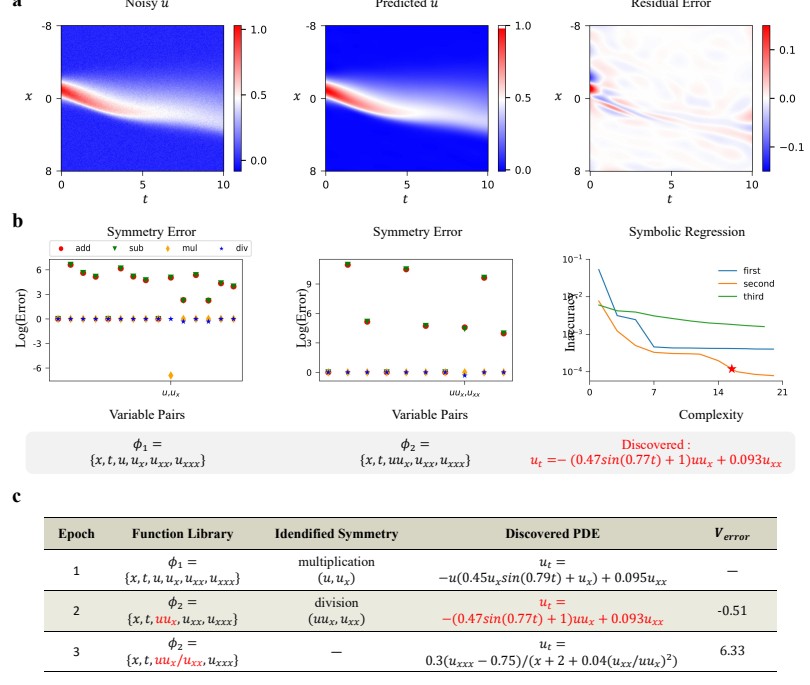

Figure S11: SI-SR results on the VC-Burgers equation. **a** Noisy input data $u$, SI-SR prediction $\hat{u}$, and residual error. **b** Symmetry identification across iterations. **c** Summary of each iteration

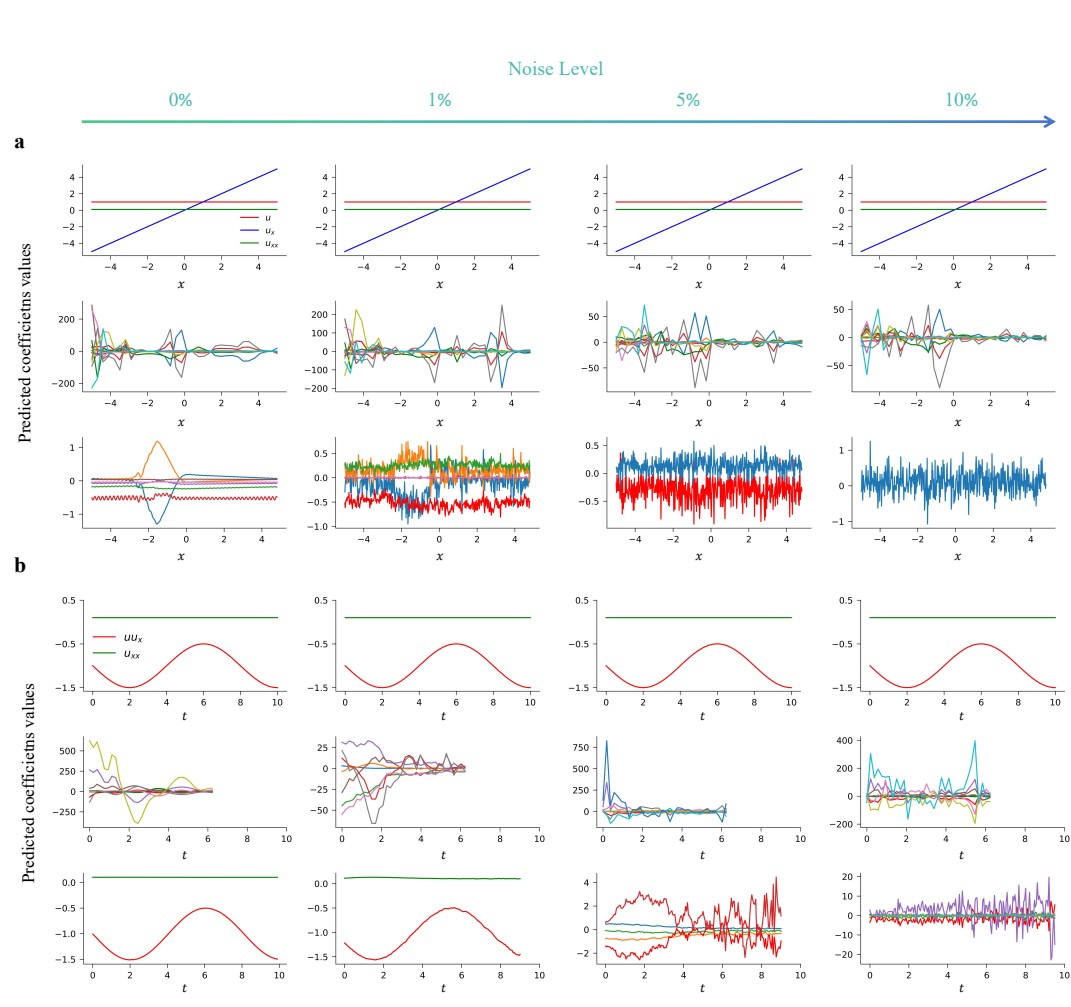

Figure S12: Comparison between SI-SR and GSTR for discovering variable coefficient PDEs. **a** Results for the spatially dependent advection–diffusion equation. The first and second columns show the results of SI-SR and GSTR, respectively, using subsampled data points. The third column shows the results of GSTR on the full dataset. **b** Results for the temporally varying Burgers' equation using the same experimental setup.

**Comparison with GSTR.** We conduct a comparative study between the proposed SI-SR framework and the GSTR method. The results are shown in Fig. S12, which illustrates the evolution of the identified coefficients with respect to time or space.

While GSTR achieves good performance when high-quality, noise-free data are available, its accuracy deteriorates substantially under noisy or data-scarce conditions. In contrast, SI-SR consistently recovers the governing equations with high accuracy across all tested scenarios. This robustness arises from the integration of automatic differentiation, which provides reliable derivative estimation, and symbolic regression, which offers flexible nonlinear modelling.

### E.3    TYPES OF NOISE

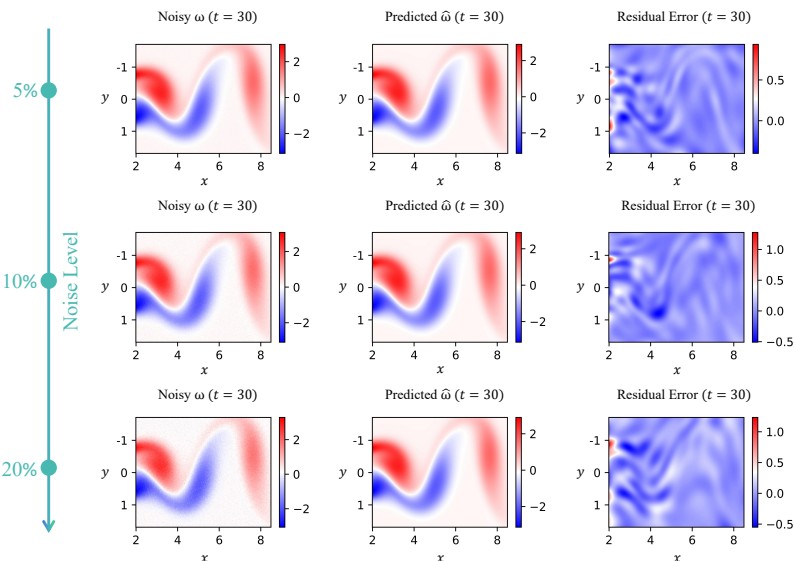

Figure S13: **The Gamma noise in the measurement data.** The noise data, predicted response, and residual error of NS equation polluted by Gamma noise.

We evaluate the robustness of SI-SR under different types of noise, including uniform and Gamma-distributed perturbations. We first examine the effect of uniform noise, defined as

$$u_{\text{noise}} = u_{\text{clean}} + (\text{noise level}) \times \text{std}(u_{\text{clean}}) \times \mathcal{U}[a, b], \qquad (46)$$

where $\mathcal{U}$ denotes a uniform distribution over the interval $[a, b]$, set to $[0, 1.7]$. Under $10\%$ uniform noise, the discovered equation is

$$\omega_t = u\omega_x + v\omega_y + 0.009956(\omega_{xx} + \omega_{yy}). \qquad (47)$$

We next consider a more challenging condition: Gamma noise, defined as

$$u_{\text{noise}} = u_{\text{clean}} + (\text{noise level}) \times \text{std}(u_{\text{clean}}) \times \mathcal{G}, \qquad (48)$$

$$p(x) = x^{k-1} \frac{e^{-x/\theta}}{\theta^k \Gamma(k)}, \qquad (49)$$

where $\mathcal{G}$ denotes the Gamma distribution, $p(x)$ is its probability density function, $k$ is the shape parameter (set to 2), $\theta$ the scale parameter (set to 0.4), and $\Gamma$ the Gamma function. Unlike uniform noise, Gamma noise is strictly positive and introduces a persistent bias, making numerical differentiation particularly challenging. Fig. S13 shows an example of measurements corrupted by Gamma noise. Under $10\%$ Gamma noise, the discovered equation is

$$\omega_t = u\omega_x + v\omega_y + 0.0098795(\omega_{xx} + \omega_{yy}). \qquad (50)$$

Although Gamma noise reduces the accuracy of automatic differentiation, SI-SR remains capable of correctly identifying the governing equation. Additional tests across varying noise levels are summarised in Table 5, which confirms that high-level Gamma noise is more difficult to handle than Gaussian or uniform noise. These results demonstrate the robustness of SI-SR against non-Gaussian noise, including distributions that are strictly positive.

## F  NUMERICAL ROBUSTNESS

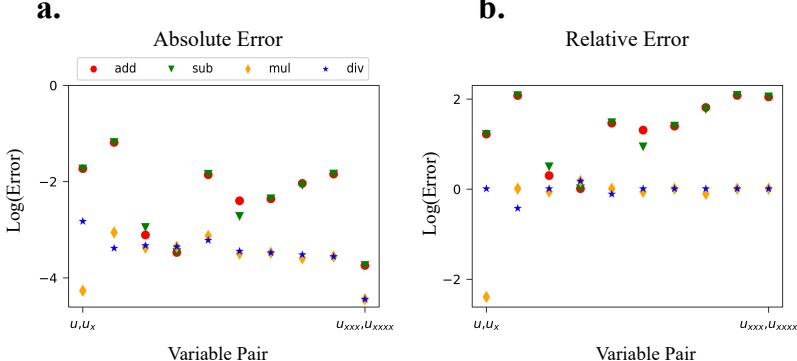

Figure S14: The symmetry error for discovering the Burgers' equation. **a** The absolute error in the symmetry method, the minimal error is division error between $u_{xxx}$ and $u_{xxxx}$. **b**: The relative error in the symmetry method, the minimal error is multiplication error between $u$ and $u_x$.

The relative error in the symmetry method is introduced to address the non-optimal function library and ensure numerical robustness. A function library is considered optimal if it contains only the necessary terms to construct the governing equation, without irrelevant or redundant elements.

Since we assume limited prior knowledge of the system—for instance, that the first-order temporal derivative of the state can be expressed in terms of the state itself and its spatial derivatives—we initialise the function library with as many candidate terms as possible. If the symmetry method is applied to such a non-optimal library, it may detect apparent symmetries that are indistinguishable from redundancies.

For example, consider the library

$$\phi_1 = \{u, u_x, u_{xx}, u_{xxx}, u_{xxxx}\},$$

constructed to discover Burgers' equation,

$$u_t = F(u, u_x, u_{xx}, u_{xxx}, u_{xxxx}) = uu_x - \gamma u_{xx}.$$

When the surrogate model correctly identifies all active terms, the four symmetry errors between the variable pair $(u_{xxx}, u_{xxxx})$ are zero, since $F$ does not depend on either $u_{xxx}$ or $u_{xxxx}$. In this case, variable pairs with zero symmetry error across all tests can be safely removed, and the remaining symmetry properties selected.

However, due to the presence of the sparse penalty term and measurement noise, sparse regression only approximates the data rather than fitting it exactly. This can be expressed in residual form:

$$\mathcal{R}_{surrogate} = \left\| \boldsymbol{U}_t - \tilde{\boldsymbol{\Phi}}\xi \right\|_2 = \left\| \boldsymbol{U}_t - \tilde{F} \right\|_2 \to 0, \tag{51}$$

where $\mathcal{R}_{surrogate}$ denotes the PDE residuals of the surrogate model. In practice, data scarcity and numerical errors in derivative computation prevent the residual from converging to zero, which can in turn affect the reliability of symmetry detection.

$\mathcal{R}_{surrogate}$ leads to symmetry errors between variables such as $(u_{xxx}, u_{xxxx})$ that are not exactly zero. To address this, we consider the *relative error* of each variable pair, rather than attempting to determine the optimal function library—a problem that remains popular yet unsolved Li et al. (2017); Chen et al. (2017); Miao & Niu (2016). An additional benefit of using relative error is improved numerical robustness.

As an illustrative example, we examine Burgers' equation ($\gamma = 1$) under $20\%$ noise with the variable set $\phi_1$. The sparse regression interpolation is expressed as:

$$\hat{F} = -0.87uu_x + 1.05u_{xx} - 0.38u^2u_{xx} + 0.16uu_{xxx} - 0.053u^2u_{xxx}. \tag{52}$$

When computing symmetry errors, the perturbations applied to variable pairs are linked to their statistical properties. For addition and subtraction, perturbations are additive and depend on the standard deviation. For multiplication and division, perturbations are multiplicative and depend on the $\ell_1$ norm. The standard deviation and $\ell_1$ norm of the candidate functions are listed in Table S7.

Due to the small $\ell_1$ norms of $u_{xxx}$ and $u_{xxxx}$, the absolute multiplication and division symmetry errors for the pair $(u_{xxx}, u_{xxxx})$ are minimal (Fig. S14a, approximately 0.00177). However, none of these exhibits a clear relative advantage over the other three types. In contrast, the relative error highlights multiplication symmetry in $(u, u_x)$ as the optimal choice, since its error is substantially lower than that of the other candidates (Fig. S14b).

Table S7: The statistical properties of the candidate function

| statistical property | $u$ | $u_x$ | $u_{xx}$ | $u_{xxx}$ | $u_{xxxx}$ |
|---|---|---|---|---|---|
| std | 0.544 | 0.170 | 0.120 | 0.132 | 0.205 |
| mean(abs) | 0.517 | 0.112 | 0.069 | 0.068 | 0.092 |

# G   DISCUSSION

Despite its advantages, the SI-SR framework also has several limitations in discovering PDEs.

First, SI-SR incurs higher computational costs compared to SINDy, mainly due to the large search space required by symbolic regression. This trade-off, however, enables the identification of variable-coefficient equations and provides an internal mechanism for validating the discovered symmetries. Such validation is particularly valuable for high-dimensional systems, where symmetry constraints play a critical role. To improve efficiency, future work could incorporate alternative approaches such as fast function extraction Vaddireddy & San (2019), deep expression trees Lample & Charton (2019), or neural symbolic methods Long et al. (2019).

Second, the choice of function library and coordinate system strongly influences whether the symmetries of the governing equations can be preserved. Since physical systems exhibit inherent invariances, representation plays a central role. However, no unified strategy currently exists for selecting coordinate systems and candidate functions Li et al. (2017); Chandrashekar & Sahin (2014). As demonstrated in our experiments, sparse regression can help redefine the function space when natural variables fail. Furthermore, broadening the scope of admissible symmetry types can reduce the burden of manual variable selection Udrescu et al. (2020). In this work, we restrict attention to four binary forms: addition, subtraction, multiplication, and division. Future extensions may incorporate more general operations, such as exponentiation or composite transformations.

## REFERENCES IN APPENDIX

Girish Chandrashekar and Ferat Sahin. A survey on feature selection methods. *Computers & electrical engineering*, 40(1):16–28, 2014.

Qi Chen, Mengjie Zhang, and Bing Xue. Feature selection to improve generalization of genetic programming for high-dimensional symbolic regression. *IEEE Transactions on Evolutionary Computation*, 21(5):792–806, 2017.

Zhao Chen, Yang Liu, and Hao Sun. Physics-informed learning of governing equations from scarce data. *Nature communications*, 12(1):6136, 2021.

Sepp Hochreiter. The vanishing gradient problem during learning recurrent neural nets and problem solutions. *International Journal of Uncertainty, Fuzziness and Knowledge-Based Systems*, 6(02): 107–116, 1998.

Diederik P Kingma. Adam: A method for stochastic optimization. *arXiv preprint arXiv:1412.6980*, 2014.

Guillaume Lample and François Charton. Deep learning for symbolic mathematics. *arXiv preprint arXiv:1912.01412*, 2019.

Jundong Li, Kewei Cheng, Suhang Wang, Fred Morstatter, Robert P Trevino, Jiliang Tang, and Huan Liu. Feature selection: A data perspective. *ACM computing surveys (CSUR)*, 50(6):1–45, 2017.

Zichao Long, Yiping Lu, and Bin Dong. Pde-net 2.0: Learning pdes from data with a numeric-symbolic hybrid deep network. *Journal of Computational Physics*, 399:108925, 2019.

Jianyu Miao and Lingfeng Niu. A survey on feature selection. *Procedia computer science*, 91: 919–926, 2016.

Samuel H Rudy, Steven L Brunton, Joshua L Proctor, and J Nathan Kutz. Data-driven discovery of partial differential equations. *Science advances*, 3(4):e1602614, 2017.

Silviu-Marian Udrescu, Andrew Tan, Jiahai Feng, Orisvaldo Neto, Tailin Wu, and Max Tegmark. Ai feynman 2.0: Pareto-optimal symbolic regression exploiting graph modularity. *Advances in Neural Information Processing Systems*, 33:4860–4871, 2020.

Harsha Vaddireddy and Omer San. Equation discovery using fast function extraction: a deterministic symbolic regression approach. *Fluids*, 4(2):111, 2019.

