# OpenReview forum: "Robust discovery of governing equations through symmetry"
_ICLR.cc/2026/Conference — ICLR 2026 Conference Withdrawn Submission_

### Official Review · Reviewer_HNWR · 2025-10-23

**Soundness:** 3
**Presentation:** 3
**Contribution:** 1
**Rating:** 2
**Confidence:** 4

**Summary:**

The paper presents a framework to identify unknown symmetries of PDE systems and use them to reduce the variable space in symbolic regression. It uses a sparse regression model as a surrogate to evaluate the symmetry loss and identify symmetries that lead to the reduction of variables by a set of specified binary operators. Another independent contribution from the use of symmetry is a model selection criterion that compares symbolic expressions with different complexity, favoring simpler solutions.

**Strengths:**

The paper presents an ambitious attempt at identifying the symmetry in a symbolic equation and discovering the equation using the symmetry at the same time. In comparison, existing methods mostly assume prior knowledge about symmetry. Then, the paper proposes new criteria for model selection and verification of the identified symmetries, though I am not sure if other model selection criteria have been developed in related works and how they compare with the proposed criteria in this paper. The paper also contains extensive experimental results, showing the method's effectiveness in a wide range of PDE systems.

**Weaknesses:**

The proposed method uses a certain class of symmetries to reduce the hypothesis space for symbolic regression. However, there are a large number of related works, some of which address more general scenarios, that this paper has not mentioned. For example, [1, 2] are the most relevant methods, which also aim to discover PDEs and utilize the much more general *Lie point symmetries* to reduce the symbolic regression problem. These papers, in my opinion, already provide a more generalized solution to the same problem compared to this submission in terms of the type of symmetries (Lie point symmetry v.s. symmetry that leads to the composition of *two* variables, as I will elaborate next), differential equations (multiple independent variables v.s. just $x$ and $t$), and base equation discovery algorithms (SINDy/genetic programming/symbolic transformer v.s. just genetic programming).

For example, to elaborate on my point on the type of symmetries, the symmetry reduction in this submission is described in the example in (5) and (6), i.e. $z_1 \odot z_2$ for a binary operator $\odot$ and two jet variables $z_1$ and $z_2$. However, not all reductions from symmetries can be done in this format. For example, consider the rotation symmetry on the 2D $(x,y)$-space. The only invariant (up to functional dependency) of this symmetry is $x^2+y^2$, meaning that any equation $F(x, y)$ with this symmetry can be expressed as $\tilde F(x^2+y^2)$. However, $x^2+y^2$ cannot be expressed by $x \odot y$ in your formulation and thus cannot be detected by the symmetry errors (7) and (8). In fact, there are abundant examples of symmetries, many of them already considered in [1,2], that the proposed method in this paper cannot apply to.

I understand that the previously mentioned papers are relatively recent, so the authors might not be aware of them. However, since this submission focuses on PDE discovery with symmetry, it is natural to consider those Lie point symmetries of PDEs. There have been many other papers with similar spirits, e.g. [3-6]. They may not have exactly targeted PDE discovery (but rather, discovering ODEs in dynamical systems, for example), but their methods are very relevant and can be easily extended to deal with partial derivatives. Without discussing these works and comparing them in terms of applicability and experimental results, it is difficult to evaluate the significance of the proposed method in this submission.

Another relatively minor comment about methodology: SINDy is used as a surrogate model for interpolating $F$ in the symmetry loss. Is it accurate enough when the ground truth dynamics contain terms that are not included in the polynomial library, e.g., trigonometric or exponential terms? On the other hand, if the ground truth equation only contains polynomial terms, why not directly use sparse regression instead of GP-based symbolic regression, since sparse regression is much more efficient in this case?

## References

[1] Discovering Symbolic Differential Equations with Symmetry Invariants. arXiv 2025.

[2] Governing Equation Discovery from Data Based on Differential Invariants. arXiv 2025.

[3] A Unified Framework to Enforce, Discover, and Promote Symmetry in Machine Learning. arXiv 2023.

[4] Symmetry-Informed Governing Equation Discovery. NeurIPS 2024.

[5] Explicit Discovery of Nonlinear Symmetries from Dynamic Data. ICML 2025.

[6] Learning fluid physics from highly turbulent data using sparse physics-informed discovery of empirical relations (SPIDER). Journal of Fluid Mechanics, 2024.

**Questions:**

* L121 "$\mathcal F (\phi_1)$ denotes nonlinear functions generated by ...": can you elaborate on which function space $\mathcal F$ exactly represents? What does "nonlinear functions" exactly mean? Does it exclude linear functions?
* Sec 3.2: Are there other notable model selection criteria for (genetic-programming-based) symbolic regression in existing works? How do they compare to the proposed criteria in your paper?
* L187: Does (12) work for most general equations? The condition is equivalent to $1 - L_{i+p} / L_i > 0.1p/d$. If $p \geq 10d$, $M_{i+p}$ can never be superior to $M_i$. However, some systems may require such complex equations to be described accurately.

---

### Official Review · Reviewer_bhAv · 2025-10-27

**Soundness:** 1
**Presentation:** 1
**Contribution:** 2
**Rating:** 2
**Confidence:** 4

**Summary:**

This work is capable of identifying and verifying symmetries from PDE systems, thereby enhancing the robustness of symbolic regression against noise. It assumes that local invariants can be derived from a set of variables through addition, subtraction, multiplication, or division, and iteratively explores and combines these operations to identify the correct symmetries. Experiments on constant and variable coefficient PDEs with varying levels of noise demonstrate that the method exhibits strong robustness.

Overall, I find the idea of iteratively searching for symmetries through combinations of existing variables to be novel. However, the manuscript is poorly articulated and lacks rigor, with many critical details omitted. Furthermore, the absence of comparisons with related work obscures the contributions of this study. Therefore, I believe this manuscript requires significant revisions.

**Strengths:**

- The idea of finding symmetries through iterative combinations of variables is novel.

- The experiments cover a variety of PDE scenarios, taking into account both constant and variable coefficient PDEs.

**Weaknesses:**

- Much earlier work (https://arxiv.org/abs/2405.16756, https://arxiv.org/abs/2505.12083) has explored symmetry-guided equation discovery, but none of them are cited or compared.

- To my understanding, this method can only recover simple and well-known symmetries (e.g., Galilean symmetry), as it assumes that all invariants are derived from addition, subtraction, multiplication, or division of variables (such as $u \odot u_x$). This strong assumption restricts it to toy experiments, preventing the utilization of more complex symmetries.

- The methodology is described vaguely and may confuse readers: How is $u \odot u_x$ derived in Equations (5) and (6)? Although the authors mention that $\odot = \times$ corresponds to Galilean symmetry, what about $\odot = \\{+, -, \div\\}$? This appears to be a specific example rather than an abstract and general methodological explanation. See Questions for more details.

- Mathematical notation is used inconsistently: In some places, $\\| \cdot \\|$ denotes a norm (Equations (3) and (4)), while in others, $| \cdot |$ is used (Equations (6) and (7)).

- The results in Table 3 and Table 5 lack error bar statistics. Reporting only a single run is inappropriate due to randomness.

- Figure 2 lacks a legend. It is unclear what the different colored lines represent.

**Questions:**

- In Equation (2), T acts on the variables (x, t, u), so how is its action on functions T(F) defined? Additionally, in Equation (4), what is the meaning of the transformation $T(\phi)$ on the variable set?

- Where do the terms $u+a$ and $u_x-a$ in Equation (7) come from? You defined $\epsilon_{1, 2, +}$, so how are the other $\epsilon_{i, j, \odot}$ defined? Why is the median used instead of the mean to calculate the error?

- Where does $\tilde{\phi}(u)$ in Equation (10) come from? Is it $\tilde{\phi}_1$ from Equation (9)?

- How are $L$, $p$, $M$, and $d$ in Equation (12) defined? Although the author explains them in natural language, their meanings remain unclear. I suggest adding brief mathematical equations to formalize their definitions.

- In Table 2 and Table 4, what is the meaning of the "identified symmetry" column (e.g., $(u, u_x, \times)$)? To my knowledge, PDE symmetries are usually expressed in the form of generators. How is your notation related to this?

---

### Official Review · Reviewer_qoHk · 2025-10-31

**Soundness:** 3
**Presentation:** 3
**Contribution:** 3
**Rating:** 8
**Confidence:** 4

**Summary:**

This paper introduces an interesting symmetry based equation discovery method. They modify sparse-regression to have a step which discovers potential symmetries and use it add or remove terms from their function bank.
In normal sparse regression we'd be finding linear combination of basis functions.
For governing equations, the functions also include derivatives, like $u_x = \partial u/\partial x$.
Their idea is to pick pairs of functions in our current basis and transform them in opposite way and check whether their current surrogate model improves or worsens as a fit to the dynamics. This could reveal whether the two functions appear together in some kind of invariant.
Then they use Lasso to weed out some basis functions and introduce new ones based on which pairs showed symmetry.

They test their method on a few system such as Burgers equation, KdV and reaction diffusion. As baseline, they compare against AI Feynman.

**Strengths:**

1. Quite intersting idea for combining symmetry with equation discovery. The pairwise invariance checking makes the search space a bit tractable.
2. Using four operations, instead of just shifts, adds an interesting dimension to the possible invariants.
3. The iterative refinement of the function library could lead to discovery of more complex invariants.
4. The method works much better than AI Feynman and is much faster.
5. It seems quite robust to noise too.

**Weaknesses:**

1. Accuracy should depend a lot on the surrogate model, but no ablation is done on it. The sine activation, number of layers etc, need at least one example of ablation studies.
2. The surrogate model may not be fitting the symmetries of the data well, resulting in poor symmetry discovery. Conversely, an equivariant surrogate would introduce bias.
3. The hope is the iterative method can find invariants beyond just the pairwise ones, but it is not clear if complex invariants, e.g. related to  special conformal transformation, would be discoverable.
4. Some related work is not discussed, e.g. Yang et al "Symmetry-informed governing equation discovery" NeurIPS 2024. They don't discover the symmetries, but do seem to force the equation to respect known symmetries.

**Questions:**

1. Was there a single run for each, or multiple? How often does the model actually succeed s fail?
2. Is the perfomance of the surrogate model reported? I know you are using early stopping, but is it the same threshold for all models?
3. If you use two very different surrogate models with the same stopping criteria, do they perform equally well for symmetry and eq discovery? What about replacing sine with tanh activation? You need some ablation results on this.

---

### Official Review · Reviewer_WtMA · 2025-10-31

**Soundness:** 2
**Presentation:** 1
**Contribution:** 2
**Rating:** 2
**Confidence:** 3

**Summary:**

This paper presents SI-SR (Symmetry-Inspired Symbolic Regression), a framework for data-driven discovery of governing equations in dynamical systems.
The method aims to improve robustness under noise and data scarcity by identifying symmetries from data and embedding them into a symmetry-constrained variable set used for symbolic regression.
A validation step is employed to confirm the detected symmetries and guide recursion.
The results on several canonical PDEs and variable-coefficient systems suggest that incorporating symmetry priors can help reduce redundancy and recover more compact representations.

**Strengths:**

- The proposed SI-SR framework demonstrates robustness under noisy and sparsely sampled data, which are common challenges in data-driven PDE discovery.

- The experimental evaluation is relatively comprehensive, covering a various levels of noise and multiple types of PDEs (Burgers’, KdV, Navier–Stokes, and reaction–diffusion systems). Additionally, the paper also reports a sensitivity analysis of SI-SR with respect to noise types and data density, providing insight into its empirical behavior across scenarios.

**Weaknesses:**

- Lack of rigorous formulation, especially in Section 3.1 (Identification of symmetry).
The core procedure for symmetry discovery is presented mainly through examples rather than a formally defined algorithmic framework.
Although Algorithm S1 in the appendix outlines the steps, it remains relatively high-level and lacks precise mathematical definitions and procedural details.

- Unclear notations and missing definitions.
Several symbols and expressions are insufficiently explained.
For example, the meaning of $a$ in Eq. (7) is not explained; the norms in Eqs. (3), (4), and (6) are undefined; the meaning of the “median” operation is ambiguous; and the definition for $\epsilon_{1,2,\odot}$ is missing (only \epsilon_{1,2,+} is shown).
In addition, the complexity $i$ in Section 3.2 is never clarified.
These omissions pose obstacles to readers.

- Inconsistency between method and experiments.
The main formulation in Eq. (1) assumes a single spatial variable x, yet the experiments on reaction–diffusion equations involve both x and y.
This mismatch raises questions about how the framework extends to higher-dimensional inputs.


- Insufficient comparison with recent methods.
There are some relevant approaches such as http://arxiv.org/abs/2307.05432v2, https://arxiv.org/abs/2310.17053v2 and https://arxiv.org/abs/2405.16756, which also address equation discovery with symmetry priors or invariant learning.
Including these methods in the comparison—or at least discussing their relation to SI-SR—would help clarify the novelty and positioning of this work.

- The restriction $i<j$ in the construction of invariant combinations implies that self-interactions are excluded, which may omit valid invariants, such as $u_x^2+u_y^2$.
This constraint could limit the method’s expressivity and completeness when discovering more complex PDE structures.

- The main text should include the explicit forms of the PDEs used in the experiments rather than referring readers only to the appendix.
This would improve readability and make the evaluation easier to follow.

**Questions:**

- Could the authors clarify why the reformulation from Eq. (3) to Eq. (4) alleviates the intractability issue? And, how is T determined in practice?

- The coefficient 0.1 in Eq. (12) appears to be empirically chosen.
Could the authors elaborate on its motivation and sensitivity?

- Criterion $V_{error} <1$.
The paper accepts symmetry substitutions when the validation error
is below 1.
How was this threshold selected?
Does it correspond to a theoretical justification or an empirical rule of thumb?

---

### Note · Authors · 2025-11-12

I have read and agree with the venue's withdrawal policy on behalf of myself and my co-authors.